# Individualized Tour Route Plan Algorithm Based on Tourist Sight Spatial Interest Field

**Xiao Zhou [1,2], Yinhu Zhan [2,\*], Guanghui Feng [3], De Zhang [4,5] and Shaomei Li [2]**

[1] Tourism Department, Leshan Vocational and Technical College, Leshan 614000, China; zhouxiao@infu.ac.cn

[2] Information Engineering University, Zhengzhou 450001, China; shaomeili@infu.ac.cn

[3] Institute of Information Engineering, Zhengzhou University of Industrial Technology, Zhengzhou 451159, China; fgh@stu.zzgyxy.edu.cn

[4] State Key Laboratory of Geo-Information Engineering, Xi'an 710054, China; zhangde01@infu.ac.cn

[5] Xi'an Research Institute of Surveying and Mapping, Xi'an 710054, China

**\*** Correspondence: yhzhan@ashn.org.cn; Tel.: +86-186-2497-1678

**Abstract:** Smart tourism is the new frontier field of the tourism research. To solve current problems of smart tourism and tourism geographic information system (GIS), individualized tour guide route plan algorithm based on tourist sight spatial interest field is set up in the study. Feature interest tourist sight extracting matrix is formed and basic modeling data is obtained from mass tourism data. Tourism groups are determined by age index. Different age group tourists have various interests; thus interest field mapping model is set up based on individual needs and interests. Random selecting algorithm for selecting interest tourist sights by smart machine is designed. The algorithm covers all tourist sights and relative data information to ensure each tourist sight could be selected equally. In the study, selected tourist sights are set as important nodes while iteration intervals and sub-iteration intervals are defined. According to the principle of proximity and completely random, motive iteration clusters and sub-clusters are formed by all tourist sight parent nodes. Tourist sight data information and geospatial information are set as quantitative indexes to calculate motive iteration values and motive iteration decision trees of each cluster are formed, and then all motive iteration values are stored in descending order in a vector. For each cluster, there is an optimal motive iteration tree and a local optimal solution. For all clusters, there is a global optimal solution. Simulation experiments are performed and results data as well as motive iteration trees are analyzed and evaluated. The evaluation results indicate that the algorithm is effective for mass tourism data mining. The final optimal tour routes planned by the smart machine are closely related to tourists' needs, interests, and habits, which are fully integrated with geospatial services. The algorithm is an effective demonstration of the application on mass tourism data mining.

**Keywords:** spatial interest field; individuality; tour guide route; motive iteration; decision tree; cluster analysis

---

## 1. Introduction

Smart tourism is the fastest growing frontier field of tourism research. The aim of smart tourism is to improve tourists' knowledge of travel destination and help them have the best travel experience [1,2]. It is also called intelligent tourism. It uses techniques of cloud computing, Internet of Things, etc., through Internet on a portable terminal to access information about tourism resources, economy, activity, and tourists, etc., and then releases relative information for tourists [3–5]. According to the information, tourists can make time schedules and plan the trip before a

vacation. The development of smart tourism will affect the tourism experience, management, service, and marketing [6,7]. The approach of smart tourism is to provide convenient and efficient service for tourists, build the frame of smart city and tourist sight, and finally improve the quality and level of tourism service [8–11]. Smart tourism development builds on tourism science. It takes advantage of cloud computing and artificial intelligence and combines GIS technology to distribute the most relevant tourism information on the Internet [12,13]. Of all the information, tourist sight classification, locations, distinguishing features, available transportation service around, traffic conditions, and accommodation fees are the most relevant. Through the human–computer interaction process, tourists can access tourism service information and get optimal decision support. Therefore, smart tourism is an interdisciplinary field of cloud computing, artificial intelligence, and GIS, etc., with a core aim of highly efficient tourism informationization [14–16] through the method of cloud computing and artificial intelligence. As its data source support is GIS service, GIS is an important prerequisite for developing smart tourism [17–19]. Before traveling, tourists usually make a plan including travel destinations, a time schedule, a route, accommodation, and so on in accordance with their needs and interests, travel availability, and budget, etc. These are the most concerning issue for tourists and the critical preconditions for tourists to obtain the best motive benefit satisfaction [20]. The quality of travel planning will directly influence tourists' experiences and perceptions during the whole trip, influence the tourists' subjective impression of the cities and destinations visited, and thus have a determined impact on the travel destination's further marketing to attract more tourists and increase the economic benefits of tourism [21–23]. The more motive benefit satisfaction tourists obtain from the whole travel process, the better their perception of the travel experience and service is, and thus, the more likely they will be to positively evaluate the travel destinations. This will lead to a positive genuine evaluation on the Internet, which will promote a travel destination's further development and economic performance[24,25]. Thus, in smart tourism supported by geospatial data and services, the aim of smart tourism is to provide tourists with smart decision support that is highly accurate, individualized, and based on tourists' needs . It can ultimately increase the motive benefit satisfaction of tourists[26,27].

Currently, decision support for smart tourism is in the early stage of development. The techniques for developing an orientation are tourism information services and data mining. Based on mass tourism and geospatial data, tourists use smart device to select tourist sights and make tour plans by themselves. Usually, there are two modes. In the first mode, depending on the tourists' subjective perception and knowledge of tourist sights, a tourist will refer to geospatial services and tourist sights information, in addition to other tourists' evaluation, and extract useful information from mass tourism data [28–30]. By analyzing and processing the information, they form an internal perception of tourist sights and, finally, make a decision about how to travel. In the second mode, various actual and online travel agencies provide a large number of tour options for tourists. However, the two modes both have some problems. First, based on subjective perception, the routes planned by tourists are usually not the optimal one because the tourists are unfamiliar with the city or tourist sights. Meanwhile, it is difficult to extract useful and valuable information from mass tourism data as deep analysis is insufficient. The type of data is usually presented textbook-style and is crowdsourced[31,32]. An insufficient planning strategy and strong subjectivity can hardly help tourists to obtain the best motive benefit. Second, tour routes provided by travel agencies are aimed to produce a profit and may not satisfy a tourist's individualized needs and interests [33–35]. Nevertheless, tourists cannot obtain the best motive benefits from a package tour. Third, mass tourism data contains high-level data and information, such that not all the information is useful and valuable. It is difficult to find valuable data and useful tourism knowledge that meets tourists' needs and interests, while this is just the most important key to obtain the best motive benefit.

As indicated by the above analysis, this study aims to solve the problems presented; namely, that scheduled tour routes cannot meet individualized needs and interests and cannot sufficiently

combine geospatial information and services. City tourist sights contain geospatial information [36,37]. We start the modeling by setting a single tourist sight as an independent object to form a feature interest tourist sight extracting matrix. A tourist sight spatial interest field model is set up based on tourists' individualized needs and interests. According to an age index, tourists are divided into three groups, elderly group, young adult group, and children group [38,39]. The mapping relationship model between tourist sight spatial interest field model and the three groups is studied and formed. By designing and developing a smart tourist sight extracting algorithm, smart motive iteration decision trees are formed, which can help tourists to select tourist sights according to their own interests, even though they are unfamiliar with the city and tourist sights. Combining this with a geospatial service [40–42], we set up a smart tour guide route plan algorithm. Tour routes planned by the algorithm are related to motive iteration values. The motive iteration values along with tour routes are arranged in descending order matrix and the optimal one is provided for tourists. The method and algorithm built in the study can provide detailed decision support and tour routes for tourists according to their needs and interests and meet their satisfaction the most [43,44]. The main content of the paper is as follows.

- First, the current research status and content regarding smart tourism are analyzed. Focusing on the problems of tour route plan, we suppose that individualized tour plans combined with geospatial services are an important means to meet the motive benefit satisfaction of tourists and maximize tourism economic benefits for destinations.
- The research data resources and spatial ranges are ensured on the basis of problem analysis. According to the tourist sight distribution and geospatial information data, a tourist sight spatial interest field model is developed and its mapping relationship model for the three age groups of tourists is formed.
- A smart algorithm is designed and developed. Tourists can get hot tourist sights via smart machine according to their needs and interests. The smart machine calculates motive iteration values and, finally, outputs tour routes for tourists.
- Simulation experiment is designed. To get the best motive benefits for tourists is the core objective, and this objective is set up by a quantitative algorithm model as the dependent hypothesis variable to find out the optimal tour routes, which is iterated by several important independent variables. The independent variables are factors and disturb factors, including critical tourist sight information and GIS service information. After ensuring tourist sight spatial interest field mapping model and selecting interested tourist sights, all the factors and disturb factors are quantified and altered according to different motive iteration clusters and trees. By iterating and outputting motive iteration values, the relative tour routes are all obtained, of which the maximum iteration value route is the optimal one for tourists, and the sub-optimal ones will also be displayed for tourists to select as there are different situations and project suggestions. Finally, experiment data is analyzed and valuable knowledge on tour route is obtained. The data and knowledge can effectively help tourists to select tourist sights, plan the whole trip, and get the best motive benefit.

## 2. Feature Interest Tourist Sight Extracting Algorithm Based on Interest Field Mapping Model

Interest tourist sights are the basic data source to make a tour route plan and calculate motive iteration values. In a smart machine, interest tourist sights are selected automatically, and the interest tendency is the key for the machine to learn and recognize tourists' needs. Thus, the feature interest tourist sight extracting algorithm based on interest field mapping model is set up first.

### 2.1. Feature Interest Tourist Sight Extracting Matrix

As to city tourism, before tourists go to an unfamiliar city, they should know about the city and tourist sights [45]. They usually select the most interested ones to visit. To obtain a data resource, the first group of definitions are discussed.

Def 1.1 Tourist sight spatial data set $P$ . Within a certain geospatial range, the set containing some popular and typical tourist sights which are selected and ensured by certain restrictive rules, is called tourist sight spatial data set $P$ .

Def 1.2 Tourist sight spatial data subset $P_r$ . Tourist sight spatial data set $P$ can be divided into some subsets according to different properties. Each subset is called tourist sight spatial data subset $P_r$ .

Def 1.3 Tourist sight spatial subset element $P_r Q_s$ . A single tourist sight element in any given tourist sight spatial data subset $\forall P_r$ is called tourist sight subset element $P_r Q_s$ .

$$
\mathbf{P} = \begin{bmatrix}
P_1 Q_1 & \cdots & P_1 Q_{\min p_r} & 0 & 0 & 0 \\
 & & P_2 Q_{p_r{}'} & 0 & 0 & 0 \\
 & & & P_3 Q_{p_r{}''} & 0 & 0 \\
 & & \cdots & & & \\
P_{\max t} Q_1 & & \cdots & & P_{\max t} Q_{\max p_r -1} & P_{\max t} Q_{\max p_r}
\end{bmatrix} \tag{1}
$$

According to the definitions and properties, we set up city tourist sight spatial data set $P$ and divide tourist sights into $t$ groups, $P = \{ P_r \mid r \in (0,t], r \in Z^+ \}$ . In which, $P_r$ is tourist sight spatial data subset, and each subset contains $p_r$ tourist sight elements, $r \in (0,t] \in Z^+$ . In any $\forall P_r$ , a single tourist sight is coded as $P_r Q_s$ , $s \in (0, p_r] \in Z^+$ . Feature interest tourist sight extracting base vector $\vec{P_r}$ is built by tourist sight spatial subset element, $\vec{P_r} = [ P_r Q_s ]$ , $r \in (0,t] \in Z^+$ , $s \in (0, p_r] \in Z^+$ .The maximum value of tourist sight classification $\max t$ and the maximum value of subset element number $\max p_r$ are used to form $\max t \times \max p_r$ dimension feature interest tourist sight extracting matrix $\mathbf{P}$ , as Formula (1). The feature interest tourist sight extracting base vector $\vec{P_r}$ is set at each matrix row and vacant element is set by data 0. Matrix $\mathbf{P}$ is the data resource for smart machine to select tourist sights.

### 2.2. Tourist Sight Spatial Interest Field Mapping Model

According to statistics for big data of various tourist sights and classifications, different tourist groups have dissimilar interest tendencies while the same group members have the similar interests [44–46]. The tourist sight spatial interest field is an intensity structure built on different interest degrees of tourist groups. According to the age index, tourists can be divided into the children group $G_1$ , $age \in (0,18] \in Z^+$ , young adult group $G_2$ , $age \in (18,44] \in Z^+$ , and the elderly group $G_3$ , $age \in (44, +\infty] \in Z^+$ . Based on nearly one year of tourism statistics data of a certain city, we set the total number of tourists paying a visit to all tourist sights of set $P$ as $n$ , in which the number of $G_i$ is $n_i$ , as in Formula (2).

$$
n = \sum_{i=1}^{\max i} n_i \tag{2}
$$

For the tourist sight spatial data subset $P_r$ , $k_{i,r}$ is the statistics number of tourists in group $G_i$ who have paid a visit to the tourist sights in $P_r$ , $k_{i,r} \in [0, n_i] \in Z^+$ . The selection of each age group for each different subset of tourist sights is independent and identically distributed [47]. The second group of definitions are discussed.

Def 2.1 Tourist sight spatial data subset visited rate $\omega_{i\cdot r}$. The ratio of visiting tourists' number in group $G_i$ on certain tourist sight spatial data subset $P_r$ to the total tourists' number of the group $G_i$ is called tourist sight spatial data subset visited rate $\omega_{i\cdot r}$, as in Formula (3). The higher the proportion is, the higher the tendency for members of the age group to visit the tourist sight subset will be.

$$\omega_{i,r} = \frac{k_{i,r}}{n_i} \tag{3}$$

Def 2.2 Interest field mapping model. The tourist sight spatial interest field is formed from tourist sight spatial data subset $P_r$ and the visited rate $\omega_{i\cdot r}$. The mapping relationship of the interest field and age group $G_i$ is called the interest field mapping model.

Def 2.3 Interest field intensity. The visited rate $\omega_{i,r}$ of age group $G_i$ on single sight spot spatial data subset $P_r$ is called the interest field intensity of subset $P_r$ on age group $G_i$.

Tourist sight spatial data subset is set as object to select tourist sights and build tourist sight spatial interest field mapping model, as shown in Figure 1. Interest field intensity histogram of tourist sight spatial data subset $P_r$ on different age groups is shown in Figure 1 (right side). Depending on interest field intensity, the smart machine can automatically recommend tourist sight spatial data subset by the input age group.

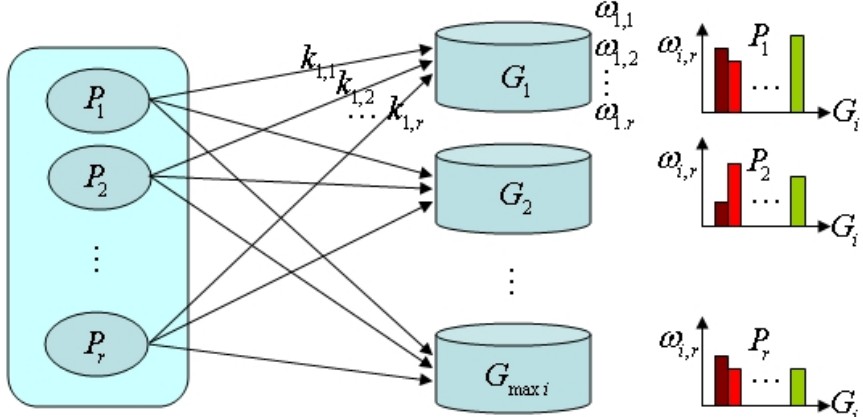

**Figure 1.** Interest field mapping model and interest field intensity histogram.

*2.3. Smart Tourist Sight Extracting Model*

According to the built feature interest tourist sight extracting matrix, tourist sight spatial interest field mapping model, and the interest field intensity histogram [48], a smart tourist sight extracting model is developed. The distribution function of the continuous random variable $X$ is set as in Formula (4).

$$F(x) = \frac{x - z_1}{z_2 - z_1} \quad x \in [z_1, z_2], 0 < z_1 < z_2 \tag{4}$$

The continuous random variable $X$ follows a uniform distribution on the interval $[z_1, z_2]$, denoted as $X \sim U(z_1, z_2)$. For any single value of x in the interval $[z_1, z_2]$, the uniform distribution function yields a finite number of elements with the maximum value in the closed interval and all the elements are random [49–52]. When the randomly selected targets are tourist sights, there is a random number interval such that $z_1 \in Z^+, z_2 \in Z^+$, and the selected random

number, $x \in Z^+$ will be subject to integer conversion. If tourists in an age group are not familiar with tourist sights, the statistics interest field intensity is used to obtain the best motive benefits. Taking one day as the travel time, the smart tourist sight extracting algorithm model is developed as follows.

**Step 1.** To make the tour route planning procedure more convenient and intelligent, tourists only need to input basic personal information and the number of tourist sights, and then the smart machine will immediately plan an optimal route. Tourists only provide their gender, their age index *age*, and the number of tourist sights to be visited. In order to have the best travel experience and avoid fatigue in one day, the number of selected tourist sight cannot exceed 5.

(1) If $m \in (5, +\infty] \cup [0] \in Z^+$, the system displays an alarm message. Review and input $m$ again. Cancel and log out of the system.

(2) If $m \in (0, 5] \in Z^+$, continue to Step.2.

**Step 2.** According to the index *age*, the smart machine ensures tourist sight spatial interest subset $P_r$ and feature interest tourist sight extracting base vector $\vec{P}_r$ in the sequence from the strongest interest field intensity to the weakest one.

**Step 3** Smart machine ensures tourist sight element number and tourist sight spatial subset elements as follows.

(1) If $m \in (0, t) \in Z^+$, arrange the $\omega_{i,r}$ related subsets in descending order. From the highest $\omega_{i,r}$ to the lowest one, one tourist sight is selected from each subset;

(2) If $m = t$, arrange $\omega_{i,r}$ related subsets in descending order. One tourist sight is selected from each subset;

(3) If $m \in (t, +\infty) \in Z^+$, arrange the $\omega_{i,r}$ related subsets in descending order. From the highest $\omega_{i,r}$ to the lowest one, one tourist sight is selected from each subset. Return, arrange the $\omega_{i,r}$ related $m - t$ subsets in descending order, and one tourist sight is selected from each subset.

**Step 4** Select a tourist sight element.

(1) If $m \in (0, t) \in Z^+$, for the first selected subset $P_r$, smart machine invokes one time of uniform distribution function on interval $(0, p_r]$ and gets $s$. Ensure there is one element tourist sight $P_r Q_s$; return, and perform the same operation on other $m - 1$ subsets; finish, and ensure that there are $m$ element tourist sights in total;

(2) If $m = t$, carry out the same operation as step (1); finish, and ensure that there are $m$ element tourist sights in total;

(3) If $m \in (t, +\infty) \in Z^+$, for the first selected subset $P_r$, smart machine invokes one time of uniform distribution function on interval $(0, p_r]$ and gets $s$. Ensure there is one element tourist sight $P_r Q_s$; return, and perform the same operation on other $t - 1$ subsets, ensure there are $t$ element tourist sights; return to the first subset $P_r$ and perform the same operation, ensure there are $m - t$ element tourist sights; finish, ensure there are $m$ element tourist sights in total.

## 3. Smart Motive Iteration Decision Tree Algorithm

Within one day, tourists visit the selected $m$ tourist sights in a single-track and non-repetitive manner. From the initial tourist sight to the last one, any single tourist sight can only be visited once. As to the selected $m$ tourist sights, there will be $A_m^m$ kinds of tour routes for the selected $m$ tourist sights, but not all the routes are optimal. Each route has a different motive benefit value for tourists, which should be measured by a quantization method. For any one tour route, a tourist starts from the first tourist sight and travels through several city roads and road nodes and arrives at the next tourist sight. In this process, the ferry distance, road traffic conditions, the convenience of GIS services, tourists' subjective perceptions, and so on will directly influence the motive benefit satisfaction of the tourist [53–55]. The motive benefit satisfaction in the previous tour interval will also influence the motive benefit satisfaction in the next tour interval. Thus, the motive iteration value of the previous tour interval is the initial value for calculating the motive iteration value of the next tour interval. Tourists travel from the first tourist sight to the last one, and in this process, the motive iteration value is monotonically increasing and thus the motive iteration function $W(T_e)$ is a monotonically increasing function. For a particular tour route, each tourist sight is related to one motive iteration function $W(T_e)$ value. For a different tour route, the same tourist sight may be related to different motive iteration function $W(T_e)$ values as the iteration process contains different values. Thus, the ultimate output motive iteration values vary for each tour route. From the aspect of the tour route selection and motive iteration process, each tour route can form a single motive iteration decision tree. There should be a maximum value of the last tourist sight, whose decision tree and related tour route is optimal and yields the highest motive benefit value for the tourist.

### 3.1. Motive Interval and Motive Sub-Interval

According to the modeling concepts, the third group of definitions are discussed.

Def 3.1 Selected tourist sight set $T$. The $m$ tourist sights selected by the smart machine which will be visited by tourists form a dataset, which is called selected tourist sight set and denoted by $T$. The element tourist sight of set $T$ is coded as $T_e$, $T = \left\{ T_e \mid e \in (0, m] \in Z^+ \right\}$.

Def 3.2 Motive iteration sub-interval $H(T_e, T_{e+1})$ and sub-interval motive iteration value $W(T_e, T_{e+1})$. Selected tourist sight set $T$ is ensured first. When tourists travel from tourist sight $T_e$ to the next one $T_{e+1}$, the tour interval between the two tourist sights, which generates motive iteration value, is called motive iteration sub-interval $H(T_e, T_{e+1})$. Within the sub-interval, an initial motive iteration value is set, and the smart machine calculates with indexes to output another motive iteration value, which is called sub-interval motive iteration value $W(T_e, T_{e+1})$. The iteration value $W(T_{e-1}, T_e)$ generated in the previous motive iteration sub-interval $H(T_e, T_{e+1})$ is the iteration function $W(T_e)$ value on tourist sight $T_e$. The value is used as the initial value to iterate the motive iteration value $W(T_{e+1})$ on tourist sight $T_{e+1}$ and it is the sub-interval motive iteration value $W(T_e, T_{e+1})$ on motive iteration sub-interval $H(T_e, T_{e+1})$. This value is also the initial value of the next sub-interval. In Figure 2, the blue interval is the motive iteration sub-interval, which is related to sub-interval motive iteration value $W(T_e, T_{e+1})$. Figure 3 shows the motive iteration function $W(T_e)$ for the motive iteration sub-interval. As seen in Figure 3, $W(T_e)$ is a discrete discontinuous and monotonically increasing function, and each tour route is related to a unique function $W(T_e)$.

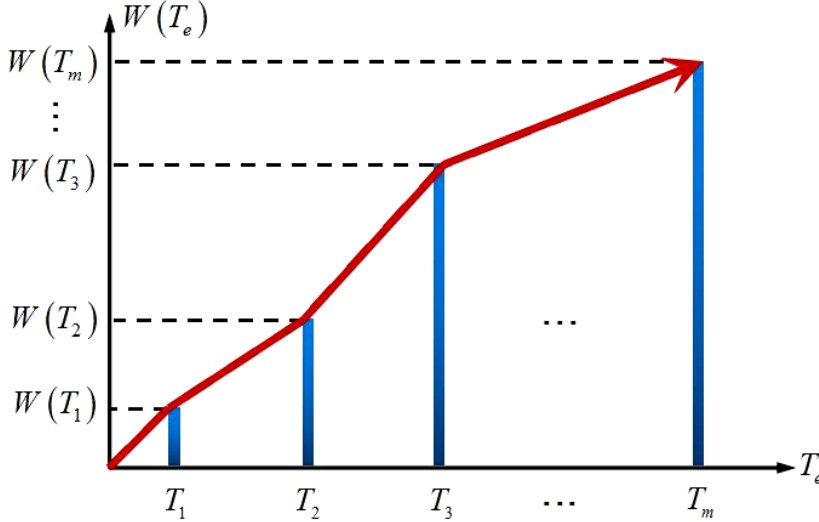

**Figure 2.** Motive iteration interval, sub-interval, and relative function values.

**Figure 3.** Motive iteration function $W(T_e)$ of a tour route.

Def 3.3 Motive iteration interval $H(T_1,T_m)$ and interval motive iteration value $W(T_1,T_m)$. The $m-1$ sub-intervals iterate from tourist sight $T_1$ to $T_m$ and, finally, output the iteration function value $W(T_m)$ on tourist sight $T_m$. The whole interval from tourist sight $T_1$ to $T_m$ is called the motive iteration interval $H(T_1,T_m)$. The iteration function value $W(T_m)$ on tourist sight $T_m$ is called the interval motive iteration value $W(T_1,T_m)$. In Figure 2, the red interval is the motive iteration interval $H(T_1,T_m)$, which is related to the interval motive iteration value $W(T_1,T_m)$.

Def 3.4 Motive iteration factor $c$. Tourists travel from tourist sight $T_e$ to $T_{e+1}$ in motive iteration sub-interval $H(T_e,T_{e+1})$ and in the process, the sub-interval motive iteration value $W(T_e,T_{e+1})$ is mainly influenced and determined by ferry distance, road traffic, the convenience of GIS services, and tourists' subjective perceptions. These factors are called motive iteration factors, $c$. The most important and decisive ones are:

- The tourist sight ferry distance $\delta_1$ (km, $\delta_1 \in (0, \max \delta_1] \in R^+$);

- Number of nearby bus line $\delta_2$ ( $\delta_2 \in [0, \max \delta_2] \in Z^+$ );
- Number of nearby subway lines $\delta_3$ ( $\delta_3 \in [0, \max \delta_3] \in Z^+$ );
- Taxi fee $\delta_4$ ( $\delta_4 \in [\min \delta_4, +\infty) \in R^+$ );
- Traffic congestion index $\delta_5$ ( $\delta_5 \in (0,1) \in R^+$ ).

Def 3.5 Motive iteration disturbance factor $\varepsilon$. In a realistic situation, each motive iteration factor has its own factors that promote or restrain itself, and these factors are called motive iteration disturbance factors, $\varepsilon$. For each factor $c$, the disturbance factor $\varepsilon$ contains the following:

- Number of road nodes $\mu_1$ ( $\mu_1 \in [0, \max \mu_1] \in Z^+$ );
- The sum of walking distance to the nearest bus station (km) and average transference time $\mu_2$ ( $\mu_2 \in (0, \max \mu_2] \in R^+$ );
- The sum of walking distance to the nearest subway station (km) and average transference time $\mu_3$ ( $\mu_3 \in (0, \max \mu_3] \in R^+$ );
- The average waiting time $\mu_4$ (h, $\mu_4 \in (0, +\infty) \in R^+$ );
- Number of roads with multiple traffic accidents, $\mu_5$ ( $\mu_5 \in [0, \max \mu_5] \in Z^+$ ). Table 1 shows the motive iteration factor $c$ and disturbance factor $\varepsilon$ values in USD in the algorithm.

Of all the factors, the average waiting time is the ratio of total waiting time to the total number of taxi stop times. During the service, a taxi may encounter traffic congestion or a red traffic light, which requires the taxi to stop and wait [56]. The total waiting time is obtained by summing all the taxi wait times.

**Table 1.** Motive iteration factors $c$ and disturbance factors $\varepsilon$ for the algorithm.

|  | $c_1$ | $c_2$ | $c_3$ | $c_4$ | $c_5$ |
|---|---|---|---|---|---|
| Motive iteration factors $c$ | $\delta_1^{-1}$ | $0.1\delta_2$ | $0.1\delta_3$ | $\delta_4^{-1}$ | $1-\delta_5$ |
|  | $\varepsilon_1$ | $\varepsilon_2$ | $\varepsilon_3$ | $\varepsilon_4$ | $\varepsilon_5$ |
| Disturbance factors $\varepsilon$ | $-0.01\mu_1$ | $-0.01\mu_2$ | $-0.01\mu_3$ | $-0.01\mu_4$ | $-0.01\mu_5$ |

We set initial motive iteration value on tourist sight $T_1$ as $W(T_1)$. Thus, the sub-interval motive iteration value $W(T_2)$ is determined by motive iteration sub-interval factor $c_{(T_1, T_2)}$, and disturbance factor $\varepsilon_{(T_1, T_2)}$ follows Formula (5).

$$W(T_2) = \sum_{u=1}^{\max u} W(T_1) \cdot c_u + \varepsilon_u \tag{5}$$

We set value $W(T_1)$, motive iteration sub-interval factor $c_{(T_1, T_2)}$, and disturbance factor $\varepsilon_{(T_1, T_2)}$. According to the calculated increase consequence due to the motive iteration function $W(T_e)$, the recursive function Formula (6) is obtained.

$$W(T_e) = \sum_{u=1}^{\max u} W(T_{e-1}) \cdot c_u + \varepsilon_u \tag{6}$$

According to Formula (6), when a tourist has finished visiting a tourist sight, a motive iteration value $W(T_{e-1})$ will be generated in the same time. This motive iteration value is the tourist's satisfaction feedback for the sub-interval. It is also the critical index to evaluate the tourist's mood, feelings, and perceptions about the travel experience, which will directly influence the satisfaction for the next sub-interval. Thus, the $W(T_{e-1})$ value is the initial motive iteration value of the next sub-interval. From the perspective of a realistic travel experience, Formula (6) accounts for the tour's objective condition and the tourist's subjective motive.

### 3.2. Motive Iteration Decision Tree Algorithm Based on M-Central Point Cluster

The selected tourist sight set $T$ elements are basic points. Apply the motive iteration algorithm to traverse all tourist sights in set $T$. Take one certain tourist sight $T_e$ of set $T$ as the central point, and other tourist sights $\neg T_e$ are points to be traversed. One central point can generate $A_{m-1}^{m-1}$ different tour routes, and each route is related to one unique motive iteration function value. The fourth group of definitions are discussed.

Def 4.1 Motive iteration decision tree $Tree_\sigma$. Tourist sight element central point $T_e$ is set as father node of decision tree. Other tourist sights $\neg T_e$ are set as child nodes. Th child nodes expansion sequence is determined by the set $T$ element display order and their permutations and combinations. One tour route generated by father node $T_e$ and its child nodes $\neg T_e$ iterations relates to one decision tree; this tree is called the motive iteration decision tree $Tree_\sigma$. One father node and $m-1$ child nodes relate to $A_{m-1}^{m-1}$ decision trees, $\sigma \in (0, A_{m-1}^{m-1}] \in Z^+$. Each decision tree relates to a unique motive iteration function $W(T_m)$ value.

Def 4.2 Decision tree child node cluster $C(\neg T_e)$. Tourist sight element central point $T_e$ is set as father node, and the other $\neg T_e$ tourist sights form a cluster, which is called decision tree child node cluster $C(\neg T_e)$ [56–58]. Figure 4 shows all the father nodes and relative child nodes which are formed by selected tourist sight set $T$ elements. Figure 4a is tourist sight distribution. Figure 4b–h is a decision tree father node and child node cluster, $m=7$, $e \in (0,8) \in Z^+$. Figure 4b shows the motive iteration decision tree formed in the way that the number one tourist sight $T_1$ is set as father node and the other $\neg T_1$ tourist sights are set as child nodes.

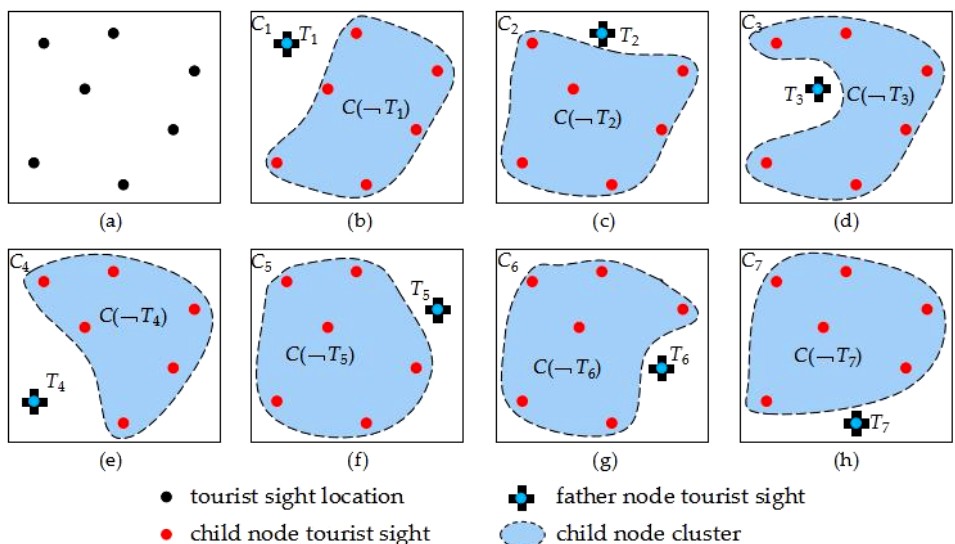

**Figure 4.** Tourist sight location, father node, and child node cluster distribution.

Def 4.3 m-central point motive iteration cluster $C_v$. As to one tourist sight element, tourist sight element $T_e$ forms $A_{m-1}^{m-1}$ motive iteration trees, and all the trees gather to one cluster, the cluster is called m-central point motive iteration cluster $C_v$, $v \in (0, m] \in Z^+$. According to the definition, selected tourist sight set $T$ contains $m$ motive iteration clusters, and each cluster contains $A_{m-1}^{m-1}$ motive iteration function $W(T_m)$ value.

Def 4.4 Motive iteration cluster local optimal solution $W[c_v]$ and global optimal solution $W[c_v]^{\max}$. According to Definition 4.2, of all the $A_{m-1}^{m-1}$ motive iteration function $W(T_m)$ values in one motive iteration cluster, the maximum value $\max W(T_m)$ is called the optimal solution of the cluster, and it is also the local optimal solution of $m$ motive iteration clusters, noted as $W[c_v]$. $m$ motive iteration clusters are related to $m$ local optimal solutions, and of all the local optimal solutions, the maximum value $\max W[c_v]$ is called global optimal solution $W[c_v]^{\max}$ of $m$ motive iteration cluster.

Def 4.5 Motive iteration cluster descending sub-vector $\vec{K}_a$ and motive iteration cluster descending vector $\vec{R}_b$. Each m-central point motive iteration cluster $C_v$ contains $A_{m-1}^{m-1}$ motive iteration decision trees and $A_{m-1}^{m-1}$ motive iteration function $W(T_m)$ values. Then, $A_{m-1}^{m-1}$ motive iteration function $W(T_m)$ values are set in descending order from the number one element location to No. $A_{m-1}^{m-1}$ one to get a descending order vector, it is called motive iteration cluster descending sub-vector $\vec{K}_{a_1,a_2}$, $a_1$ is vector code, $a_1 \in (0, m] \in Z^+$, $a_2$ is vector element location code, and $a_2 \in (0, A_{m-1}^{m-1}] \in Z^+$. According to the definition, motive iteration cluster local optimal solution $W[c_v]$ is stored in the cluster's number one element location of the motive iteration cluster descending sub-vector $\vec{K}_{a_1,a_2}$. Then, $m$ motive iteration cluster local optimal solutions are set in descending order from number one element location to number $m$ one to get a descending order vector; it is called motive iteration cluster descending vector $\vec{R}_b$, in which $b$ is vector element location code, $b \in (0, m] \in Z^+$. According to the definition, the motive iteration cluster global optimal solution $W[c_v]^{\max}$ is stored in the number one element location of motive iteration cluster descending vector $\vec{R}_b$.

According to the fourth group of definitions, we take selected tourist sight set $T$ containing $m$ tourist sights as basic data set. Each tourist sight sub-interval motive iteration factors $c(T_1, T_2)$ and disturb factors $\varepsilon(T_1, T_2)$ are ensured. Motive iteration decision tree algorithm based on m-central point cluster is designed and developed.

**Step 1.** Empty vector $\vec{K}_{a_1,a_2}$ and $\vec{R}_b$ are built. $m$ empty vectors $\vec{K}_{a_1,a_2}$ and one empty vector $\vec{R}_b$ are built. Vector $\vec{K}_{a_1,a_2}$ contains $A_{m-1}^{m-1}$ element locations. Vector $\vec{R}_b$ contains $m$ element locations.

**Step 2.** m-central point motive iteration cluster $C_v$ of the number one tourist sight element is built. The number one element location tourist sight of selected tourist sight set $T$ is taken to build

m-central point motive iteration cluster $C_1$, which contains $A_{m-1}^{m-1}$ decision trees. Decision trees are generated by father node $T_1$ and child node cluster $C\left(\neg T_1\right)$.

(1) The first decision tree is built. The first decision tree starts from tourist sight $T_1$, and traverses remaining $\neg T_1$ tourist sights to form the first tour route;

(2) Initial motive iteration function value $W\left(T_1\right)$ is set. According to Formulas (5) and (6), the smart machine outputs the first decision tree's motive iteration function $W\left(T_m\right)$ value, noted as $W\left(T_m^1\right)$;

(3) $W\left(T_m^1\right)$ is stored into the number one element location $K_{1,1}$ of vector $\vec{K}_{1,a_2}$;

(4) The second decision tree is built. The second decision tree starts from tourist sight $T_1$, and traverses remaining $\neg T_1$ tourist sights to form the second tour route;

(5) Take step (2) initial value $W\left(T_1\right)$ to iterate and output the second decision tree's motive iteration function $W\left(T_m\right)$ value, noted as $W\left(T_m^2\right)$;

(6) Compare $W\left(T_m^2\right)$ and $W\left(T_m^1\right)$. If $W\left(T_m^2\right)>W\left(T_m^1\right)$, the smart machine clears the number one element location $K_{1,1}$, stores $W\left(T_m^2\right)$ into the number one element location $K_{1,1}$ of vector $\vec{K}_{1,a_2}$, descends $W\left(T_m^1\right)$, and stores it into number two element location $K_{1,2}$ of vector $\vec{K}_{1,a_2}$; If $W\left(T_m^2\right)\leq W\left(T_m^1\right)$, smart machine keeps $W\left(T_m^1\right)$ in the number one element location $K_{1,1}$ of vector $\vec{K}_{1,a_2}$, and stores $W\left(T_m^2\right)$ into the number two element location $K_{1,2}$ of vector $\vec{K}_{1,a_2}$;

(7) Return to step (4). The third decision tree is built and $W\left(T_m^3\right)$ is obtained;

(8) Return to step (6), compare $W\left(T_m^3\right)$ and other iteration values;

(I) If $W\left(T_m^2\right)>W\left(T_m^1\right)$

① $W\left(T_m^3\right)>W\left(T_m^2\right)$, the smart machine clears the number one element location $K_{1,1}$ and the number two element location $K_{1,2}$, stores $W\left(T_m^3\right)$ into the number one element location $K_{1,1}$ of vector $\vec{K}_{1,a_2}$, and stores $W\left(T_m^2\right)$ and $W\left(T_m^1\right)$ into the number two element location $K_{1,2}$ and number three element location $K_{1,3}$ of vector $\vec{K}_{1,a_2}$ respectively;

② $W\left(T_m^2\right)\geq W\left(T_m^3\right)>W\left(T_m^1\right)$, the smart machine clears the number two element location $K_{1,2}$, keeps $W\left(T_m^2\right)$ in the number one element location $K_{1,1}$ of vector $\vec{K}_{1,a_2}$, stores $W\left(T_m^3\right)$ into the number two element location $K_{1,2}$ of vector $\vec{K}_{1,a_2}$, and stores $W\left(T_m^1\right)$ into the number three element location $K_{1,3}$ of vector $\vec{K}_{1,a_2}$;

③ $W\left(T_m^1\right) \geq W\left(T_m^3\right)$, the smart machine keeps the number one and number two element locations, and stores $W\left(T_m^3\right)$ into the number three element location $K_{1,3}$ of vector $\vec{K}_{1,a_2}$;

(II) $W\left(T_m^2\right) \leq W\left(T_m^1\right)$, the method to compare and renew element location is the same as previous step (I);

Repeat sub steps (4)-(8), and motive iteration cluster descending sub-vector $\vec{K}_{1,a_2}$ is obtained. The number one element location $K_{1,1}$ is related to local optimal solution $W\left[c_1\right]$ of motive iteration cluster $C_1$.

**Step 3.** Local optimal solution $W\left[c_1\right]$ of motive iteration cluster $C_1$ is stored into the number one element location $R_1$ of vector $\vec{R}_b$.

**Step 4.** Other motive iteration clusters and local optimal solutions are generated.

(1) Perform Step 2 sub step (1)-sub step (9); m-central point motive iteration cluster $C_2$ of the number two tourist sight element is built. Motive iteration cluster descending sub-vector $\vec{K}_{2,a_2}$ and local optimal solution $W\left[c_2\right]$ of motive iteration cluster $C_2$ are built, too;

(2) Compare $W\left[c_1\right]$ and $W\left[c_2\right]$. If $W\left[c_2\right] > W\left[c_1\right]$, the smart machine clears the number one element location $R_1$ of vector $\vec{R}_b$, stores $W\left[c_2\right]$ into the number one element location $R_1$, stores $W\left[c_1\right]$ into the number two element location $R_2$; If $W\left[c_2\right] \leq W\left[c_1\right]$, the smart machine keeps $W\left[c_1\right]$ in the number one element location $R_1$ of vector $\vec{R}_b$.

(3) Then m-central point motive iteration cluster $C_3$ of the number three tourist sight element is built. Motive iteration cluster descending sub-vector $\vec{K}_{3,a_2}$ and local optimal solution $W\left[c_3\right]$ of motive iteration cluster $C_3$ are built, too;

(4) Compare $W\left[c_3\right]$ with other local optimal solutions.

(I) $W\left[c_2\right] > W\left[c_1\right]$

① $W\left[c_3\right] > W\left[c_2\right]$, the smart machine clears the number one element location $R_1$ and the number two element location $R_2$, stores $W\left[c_3\right]$ into the number one element location $R_1$ of vector $\vec{R}_b$, and stores $W\left[c_2\right]$ into the number two element location $R_2$ and the number three element location $R_3$ of vector $\vec{R}_b$ respectively;

② $W\left[c_2\right] \geq W\left[c_3\right] > W\left[c_1\right]$, the smart machine clears the number two element location $R_2$, keeps $W\left[c_2\right]$ in the number one element location $R_1$ of vector $\vec{R}_b$, stores $W\left[c_3\right]$ into the number two element location $R_2$ of vector $\vec{R}_b$, and stores $W\left[c_1\right]$ into the number three element location $R_3$ of vector $\vec{R}_b$;

③ $W[c_1] \geq W[c_3]$, the smart machine keeps the number one and number two element location, and stores $W[c_3]$ into the number three element location $R_3$ of vector $\vec{R_b}$;

(II) $W[c_2] \leq W[c_1]$, the method to compare and renew element location is the same as previous step (I);

**Step 5.** Repeat Step 4 sub step (1)-sub step (4) until $m$ motive iteration cluster descending sub-vectors and relative local optimal solutions are obtained. Output the motive iteration cluster descending vector $\vec{R_b}$. The number one element location is the motive iteration cluster global optimal solution $W[c_v]^{max}$.

According to Step 1~Step 5, the Algorithm 1 pseudo-code is as follows.

---

**Algorithm 1.** The algorithm to generate $\vec{R_b}$ and $W[c_v]^{max}$

1:  **Set** $\vec{K}_{a_1,a_2}$ and $\vec{R_b}$. $a_1 \in (0, m] \in Z^+$, $a_2 \in (0, A_{m-1}^{m-1}] \in Z^+$, $b \in (0, m] \in Z^+$;

2:  As to $C_v$: **For** $v=1$, $v{++}$, $v \leq m$ and **For** $b=1$, $b=b+1$, $b \leq m$

3:  As to $Tree_\sigma$: **For** $\sigma=1$, $\sigma=\sigma+1$, $\sigma \leq A_{m-1}^{m-1}$ and **For** $a_1=1$, $a_1=a_1+1$, $a_1 \leq m$

4:  **Output** $W(T_m^\sigma)$;

5:  **Compare** $W(T_m^\sigma)$ and $W(T_m^{\sigma'})$;

6:  **Array** $\vec{K}_{a_1,a_2}$ in descending order from $W(T_m^1)$ to $W(T_m^{A_{m-1}^{m-1}})$;

7:  **Output** $\vec{K}_{a_1,a_2}$ and $W[c_v]$;

8:  **Array** $W[c_v]$ in descending order from $W[c_1]$ to $W[c_2]$;

9:  **Output** $\vec{R_b}$ and $W[c_v]^{max}$

10: **End procedure**

---

The motive iteration cluster location optimal solution $W[c_v]$ and the global optimal solution $W[c_v]^{max}$ have practical value as smart machine has considered tourists' individualized needs and interests.

**Situation one:** The principle of proximity.

After a tourist or smart machine selects tourist sights, the tourist may consider taking the whole trip starting with the nearest tourist sight, as their temporary accommodation may be close to the first tourist sight. Thus, the principle of proximity states that, the nearest tourist sight is the starting point; or if the tourist is particularly interested in a certain tourist sight, he may wish to visit that particular one first [27]. In this situation, the smart machine could only consider the decision tree, cluster, and its motive iteration cluster local optimal solution generated by tourist sight visited first.

**Situation two:** The principle of completely random.

After a tourist or smart machine selects tourist sights, the tourist may have no particular requirement on the tour route, and they will accept any provided tour route [45–47]. In this situation, the smart machine should consider motive iteration cluster global optimal solution and relative tour route first in order to best meet tourists' best motive benefits and needs.

## 4. Example Simulation Experiment and Data Analysis

The research range is one particular downtown area of a city; all the factors, including disturbance factors, come from urban GIS services. The routes planned in the study are all based on city urban roads and avenues. The tourist sights selected for the original data source are all urban

tourist sights which are located in the city but not in the outskirts. In one city, the factors and disturbance factors mentioned in the study are identical, and it is appropriate to use the identical factors as parameters to do the study, as different cities have different conditions. Thus, the algorithm presented in the study is suitable for a city urban tour. Taking Zhengzhou city as the data resource of tourist sight and GIS services, the basic data were sampled before the performance of example simulation experiment [2].

### 4.1. Data Sampling

In terms of Zhengzhou city's urban tourist sights and GIS services, the data sampling range and objects should meet the following conditions [50–53]. (1) The research range is continuous in geographic space, and it covers all the city's main districts. (2) The tourist sights should be representative and have a steady visitor flow volume, be well-equipped, and fully functional [56–59]. Commonly, they have a geographical advantage and convenient city service. (3) The tourist sights are independent, and do not influence each other with respect to the tour. (4) They are closely connected by a convenient urban road network. Tourists can go back and forth between two arbitrary tourist sights. According to the standard, Zhengzhou city's third ring road, six north–south roads, and seven east–west roads are selected to form the geographic research range. Within the range, 27 tourist sights are selected as study objects. Tourist sight spatial dataset $P$, tourist sight spatial data subset $P_r$, and tourist sight spatial subset element $P_r Q_s$ are constructed.

According to the first group of definition and the different properties, the urban tourist sight spatial dataset $P$ contains $P_1$, $P_2$, $P_3$ and $P_4$ four subsets. $P_1$ is the park and green land set. $P_2$ is the amusement place set. $P_3$ is the venue set. $P_4$ is the shopping center set. The sampling data are reported in Table 2 and the tourist sight distribution is shown in Figure 5.

**Table 2.** Tourist sight spatial datasets, subsets, and elements.

| Dataset | $P$ | | | |
|---|---|---|---|---|
| **Data Subset** | $P_1$ | $P_2$ | $P_3$ | $P_4$ |
| Element | $P_1Q_1$ Botanical Park<br>$P_1Q_2$ Bishagang Park<br>$P_1Q_3$ Renmin Park<br>$P_1Q_4$ Zijingshan Park<br>$P_1Q_5$ Lvcheng Square<br>$P_1Q_6$ Forest Park<br>$P_1Q_7$ Zhengzhou Zoo<br>$P_1Q_8$ Yueji Park<br>$P_1Q_9$ Xiliuhu Park | $P_2Q_1$ Century Park<br>$P_2Q_2$ Water fun Park<br>$P_2Q_3$ Children Park<br>$P_2Q_4$ Fun street | $P_3Q_1$ Henan museum<br>$P_3Q_2$ City museum<br>$P_3Q_3$ Science museum<br>$P_3Q_4$ Erqi memorial<br>$P_3Q_5$ Zhongyuan tower<br>$P_3Q_6$ Aquarium | $P_4Q_1$ Wangfujing<br>$P_4Q_2$ Erqi Wanda<br>$P_4Q_3$ Zhongyuan Wanda<br>$P_4Q_4$ Guomao 360<br>$P_4Q_5$ CC mall<br>$P_4Q_6$ Dehua street<br>$P_4Q_7$ Dashanghai<br>$P_4Q_8$ Dennis |

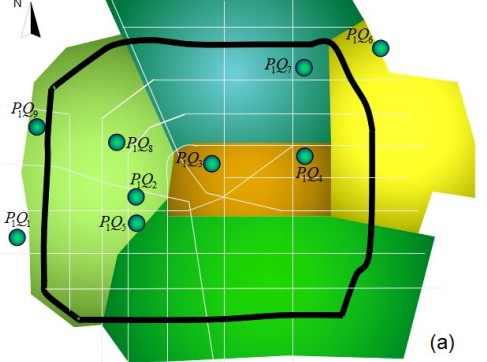

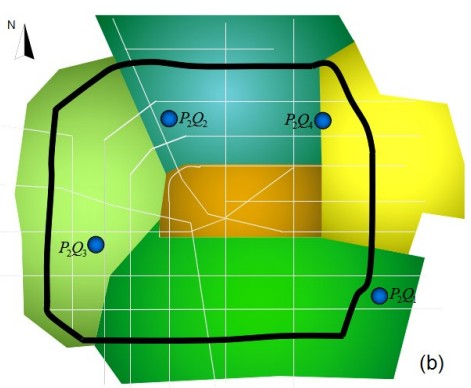



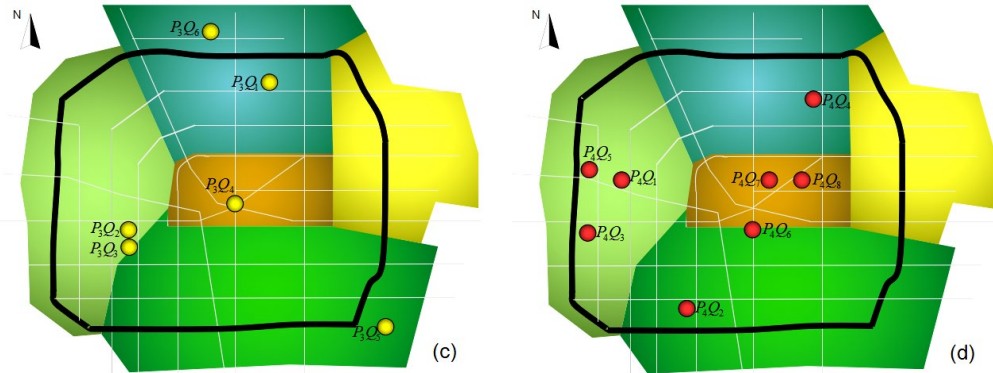

**Figure 5.** Tourist sight spatial data subset element distribution. Panels (**a**), (**b**), (**c**) and (**d**) are the distributions of tourist sight spatial data subsets $P_1 \sim P_4$. Each figure contains Zhengzhou city's five main districts and city arterial roads. Grey roads are a north–south or east–west orientation, whereas the pitch-black road is the third ring road.

According to the data, the $4 \times 9$ dimension feature interest tourist sight extracting matrix $\mathbf{P}$ is defined in Formula (7).

$$\mathbf{P} = \begin{bmatrix} P_1Q_1 & P_1Q_2 & P_1Q_3 & P_1Q_4 & P_1Q_5 & P_1Q_6 & P_1Q_7 & P_1Q_8 & P_1Q_9 \\ P_2Q_1 & P_2Q_2 & P_2Q_3 & P_2Q_4 & 0 & 0 & 0 & 0 & 0 \\ P_3Q_1 & P_3Q_2 & P_3Q_3 & P_3Q_4 & P_3Q_5 & P_3Q_6 & 0 & 0 & 0 \\ P_4Q_1 & P_4Q_2 & P_4Q_3 & P_4Q_4 & P_4Q_5 & P_4Q_6 & P_4Q_7 & P_4Q_8 & 0 \end{bmatrix} \tag{7}$$

According to the second group of definition, tourist sight spatial interest field mapping model of Zhengzhou city is set up. In terms of tourist statistics, each group's number of people is 350, $n_i = 350$, $i \in (0, 4) \in Z^+$. The visiting tourists and the visited rate for each tourist sight subset are reported in Table 3. From Table 3, the interest field intensity for each group for each tourist sight can be analyzed, and visual graphs are obtained. Elderly people have the greatest interest in park and green land subset $P_1$, followed by children, while young adults have the least interest. Children have the greatest interest in amusement place set $P_2$, followed by young adults, while elderly people have the least interest. All groups have relatively identical interest in venue set $P_3$. Young adults have the greatest interest in shopping center set $P_4$, followed by children, while elderly people have the least interest. According to the interest field mapping model and intensity graphs, the smart machine will determine the interest tendencies and provide the proper tourist sights after the tourist input the number of tourist sight to visit.

**Table 3.** The visiting tourists and visited rate for each group and tourist sight subset.

| | | $n_1(G_1)$ | $n_2(G_2)$ | $n_3(G_3)$ | $k_{1,r}$ | $k_{2,r}$ | $k_{3,r}$ | $\omega_{1,r}$ | $\omega_{2,r}$ | $\omega_{3,r}$ |
|---|---|---|---|---|---|---|---|---|---|---|
| $P_1$ | $r=1$ | | | | 223 | 128 | 311 | 0.637 | 0.366 | 0.889 |
| $P_2$ | $r=2$ | 350 | 350 | 350 | 309 | 145 | 89 | 0.883 | 0.414 | 0.254 |
| $P_3$ | $r=3$ | | | | 288 | 269 | 264 | 0.823 | 0.769 | 0.754 |
| $P_4$ | $r=4$ | | | | 190 | 337 | 145 | 0.543 | 0.963 | 0.414 |

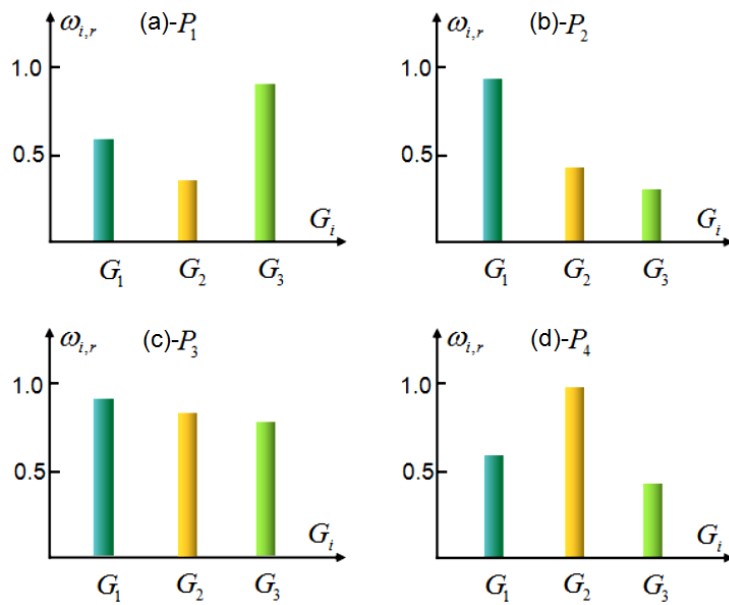

**Figure 6.** Each tourist sight's interest field intensity for each age group. (**a**) is the park and green land subset, (**b**) is the amusement place set, (**c**) is the venue set , and (**d**) is the shopping center set.

*4.2. Simulation Experiment and Results Analysis*

Based on the algorithm models developed and the data sampling, a simulation experiment is performed, and data result analysis is obtained.

4.2.1. Simulation Experiment

A young adult plans to have a trip in downtown of Zhengzhou city, but he is not familiar with the city. He wishes to visit four tourist sights within one day. According to his requirements, the smart machine plans a tour route for him. First, referring to the interest field mapping model and intensity, the smart machine matches the background data and determines that this tourist belongs to the young adult group and the machine determines that he may have a high level of interest tendency in tourist sights in the shopping set and venue subset but a low level of interest tendency in park and green land park. Thus, the most interested tourist sights are at the top of the priority list for the smart machine to select. Meanwhile, tourist sight repeat ability should be set to avoid the same tourist sights being selected. Meanwhile, less interested tourist sights are also considered but the least interested ones are avoided to ensure that the tourist sight selection is comprehensive and diverse. Matrix $\mathbf{P}$ is used as a data resource from which to extract the proper tourist sights. In the simulation experiment, the smart machine selects and recommends the following tourist sights.

- Sample 1：$1\text{-}P_1$ , $2\text{-}P_3$ , $1\text{-}P_4$ ;
- Sample 2: $1\text{-}P_1$ , $2\text{-}P_4$ , $1\text{-}P_3$ ;
- Sample 3: $1\text{-}P_2$ , $2\text{-}P_3$ , $1\text{-}P_4$ ;
- Sample 4: $1\text{-}P_2$ , $2\text{-}P_4$ , $1\text{-}P_3$ ;
- Sample 5: $2\text{-}P_3$ , $2\text{-}P_4$ .

The young adult tourist selects one of the five samples according to his own interests and needs. If he has no particular preference of the recommended samples, the smart machine will randomly select one sample for him. Take Sample 1, for instance. The smart machine provides him with one famous park, two famous venues, and one shopping mall; they are $P_1Q_2$ Bishagang park, $P_3Q_4$

Erqi memorial, $P_3Q_1$ Henan museum, and $P_4Q_3$ Zhongyuan Wanda. Store the selected tourist sight into matrix $T$, as in Formula (7).

$$T=\{T_1:P_1Q_2,T_2:P_3Q_4,T_3:P_3Q_1,T_4:P_4Q_3\} \qquad (8)$$

Tourist sights $T_1$, $T_2$, $T_3$ and $T_4$ are taken as father nodes, respectively, and the decision tree child node clusters $C(\neg T_e)$ and the motive iteration clusters $C_v$ are built as shown in Figure 7.

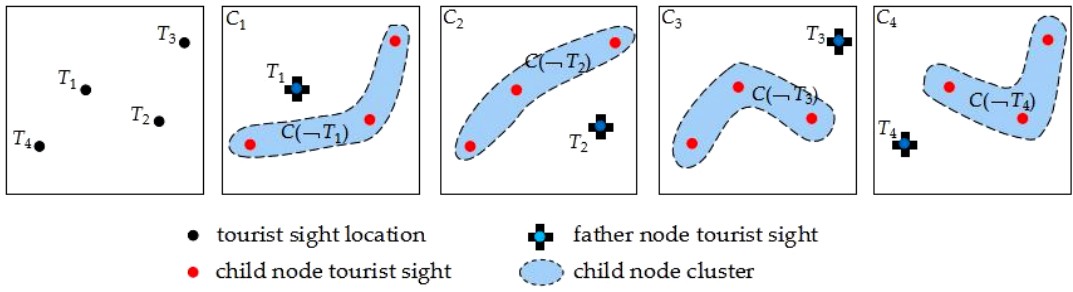

**Figure 7.** Building the decision tree child node clusters $C(\neg T_e)$ and the motive iteration clusters $C_v$.

Tourist sight $T_1$ is taken as father node to build the decision tree and six trees are formed in total. Each decision tree contains three motive iteration sub-intervals $H(T_e,T_{e+1})$, the corresponding three sub-interval motive iteration values $W(T_e,T_{e+1})$, one motive iteration interval $H(T_1,T_m)$, and one interval motive iteration value $W(T_1,T_m)$. Each decision tree and sub-interval relates to the motive iteration function $W(T_e)$ and function value, and the maximum value of motive iteration function $W(T_e)$ relates to motive iteration cluster local optimal solution $W[c_v]$. Regarding the other tourist sights $T_2$, $T_3$ and $T_4$, the method to generate the decision tree and local optimal solution is identical. From the basic GIS service data for Zhengzhou city, the motive iteration factor $c$ and disturbance factor $\varepsilon$ of each sub-interval are constructed, as Table 4 and Figure 8 indicates. In Figure 8, the abscissa is $c$ and the ordinate is $-\varepsilon$. According to the algorithm in the third segment, each decision tree sub-interval motive iteration value $W(T_e,T_{e+1})$, interval motive iteration value $W(T_1,T_m)$, motive iteration function $W(T_m)$ value, and motive iteration cluster local optimal solution $W[c_v]$ are obtained; the initial values are all $W(T_1)=1.000$, as Table 5 shows. Each decision tree's sub-interval motive iteration values and interval motive iteration values are shown in Figures 9 and 10.

**Table 4.** The tourist sight motive iteration factors, $c$ and disturbance factors, $\varepsilon$ for the simulation experiment.

| Sub-Interval | $c_1$ ($\varepsilon_1$) | $c_2$ ($\varepsilon_2$) | $c_3$ ($\varepsilon_3$) | $c_4$ ($\varepsilon_4$) | $c_5$ ($\varepsilon_5$) |
|---|---|---|---|---|---|
| $T_1,T_2(T_2,T_1)$ | 0.233 (−0.040) | 0.400 (−0.012) | 0.100 (−0.006) | 0.083 (−0.002) | 0.391 (−0.010) |
| $T_1,T_3(T_3,T_1)$ | 0.123 (−0.110) | 0.300 (−0.022) | 0.100 (−0.018) | 0.050 (−0.003) | 0.254 (−0.020) |
| $T_1,T_4(T_4,T_1)$ | 0.345 (−0.080) | 0.400 (−0.012) | 0.100 (−0.015) | 0.100 (−0.002) | 0.549 (−0.010) |
| $T_2,T_3(T_3,T_2)$ | 0.175 (−0.120) | 0.400 (−0.024) | 0.100 (−0.024) | 0.067 (−0.003) | 0.412 (−0.020) |
| $T_2,T_4(T_4,T_2)$ | 0.141 (−0.150) | 0.400 (−0.014) | 0.100 (−0.019) | 0.059 (−0.003) | 0.502 (−0.010) |
| $T_3,T_4(T_4,T_3)$ | 0.097 (−0.090) | 0.400 (−0.022) | 0.100 (−0.031) | 0.048 (−0.004) | 0.212 (−0.020) |

**Table 5.** Decision tree interval motive iteration values and global optimal solution.

| Father Node | $\sigma$ | Decision Tree | $W\left(T_e, T_{e+1}\right)$ | | | | $W\left(T_m\right)$ | | | $W\left[C_v\right]$ |
|---|---|---|---|---|---|---|---|---|---|---|
| $T_1$ | 1 | $T\ \ T\ T\ T$ | 1.137 | 1.121 | 0.794 | 1.000 | 2.137 | 3.258 | 4.052 | |
| | 2 | $T\ \ T\ T\ T$ | 1.137 | 1.171 | 0.837 | 1.000 | 2.137 | 3.308 | 4.145 | |
| | 3 | $T\ \ T\ T\ T$ | 0.654 | 0.564 | 0.482 | 1.000 | 1.654 | 2.218 | 2.700 | 5.322 |
| | 4 | $T\ \_T\ T\ T$ | 0.654 | 0.393 | 0.276 | 1.000 | 1.654 | 2.047 | 2.323 | |
| | 5 | $T\ \ T\ T\ T$ | 1.375 | 1.457 | 1.490 | 1.000 | 2.375 | 3.832 | 5.322 | |
| | 6 | $T\ \ T\ T\ T$ | 1.375 | 1.011 | 0.976 | 1.000 | 2.375 | 3.386 | 4.362 | |
| $T_2$ | 1 | $T\ \ T\ T\ T$ | 1.137 | 0.767 | 0.490 | 1.000 | 2.137 | 2.904 | 3.394 | |
| | 2 | $T\ \ T\ T\ T$ | 1.137 | 1.580 | 1.187 | 1.000 | 2.137 | 3.717 | 4.904 | |
| | 3 | $T\ \_T\ T\ T$ | 0.963 | 0.623 | 0.812 | 1.000 | 1.963 | 2.586 | 3.393 | 4.904 |
| | 4 | $T\ \_T\ T\ T$ | 0.963 | 0.658 | 0.864 | 1.000 | 1.963 | 2.621 | 3.485 | |
| | 5 | $T\ \_T\ T\ T$ | 1.006 | 1.384 | 0.972 | 1.000 | 2.006 | 3.390 | 4.362 | |
| | 6 | $T\ \ T\ T\ T$ | 1.006 | 0.695 | 0.402 | 1.000 | 2.006 | 2.701 | 3.103 | |
| $T_3$ | 1 | $T\ \ T\ T\ T$ | 0.654 | 0.719 | 0.668 | 1.000 | 1.654 | 2.373 | 3.041 | |
| | 2 | $T\ \ T\ T\ T$ | 0.654 | 0.858 | 0.835 | 1.000 | 1.654 | 2.512 | 3.347 | |
| | 3 | $T\ \ T\ T\ T$ | 0.963 | 1.092 | 1.512 | 1.000 | 1.963 | 3.055 | 4.567 | 4.567 |
| | 4 | $T\ \ T\ T\ T$ | 0.963 | 0.962 | 1.318 | 1.000 | 1.963 | 2.925 | 4.243 | |
| | 5 | $T\ \ T\ T\ T$ | 0.690 | 0.912 | 1.031 | 1.000 | 1.690 | 2.602 | 3.633 | |
| | 6 | $T\ \ T\ T\ T$ | 0.690 | 0.633 | 0.694 | 1.000 | 1.690 | 2.323 | 3.017 | |
| $T_4$ | 1 | $T\ \ T\ T\ T$ | 1.375 | 1.590 | 2.256 | 1.000 | 2.375 | 3.965 | 6.221 | |
| | 2 | $T\ \ T\ T\ T$ | 1.375 | 0.964 | 0.921 | 1.000 | 2.375 | 3.339 | 4.260 | |
| | 3 | $T\ \ T\ T\ T$ | 1.006 | 1.144 | 0.773 | 1.000 | 2.006 | 3.150 | 3.923 | 6.221 |
| | 4 | $T\ \ T\ T\ T$ | 1.006 | 0.970 | 0.629 | 1.000 | 2.006 | 2.976 | 3.605 | |
| | 5 | $T\ \ T\ T\ T$ | 0.690 | 0.398 | 0.410 | 1.000 | 1.690 | 2.088 | 2.498 | |
| | 6 | $T\ \ T\ T\ T$ | 0.690 | 0.605 | 0.660 | 1.000 | 1.690 | 2.295 | 2.955 | |

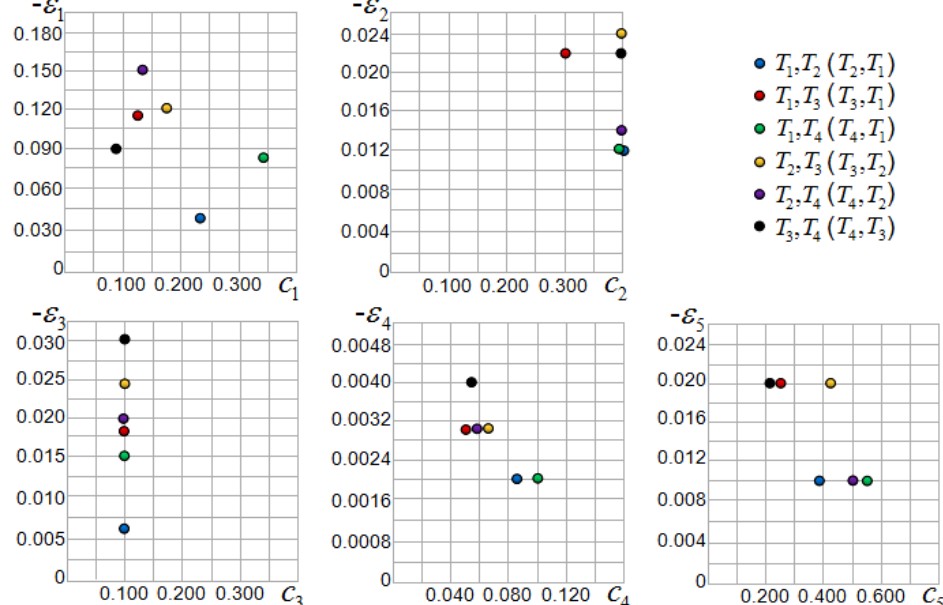

**Figure 8.** Distribution of the simulation experiment tourist sight motive iteration factor $C$ and disturbance factor $\varepsilon$. To display all the data in the first quadrant, the ordinate value is set as $-\varepsilon$.

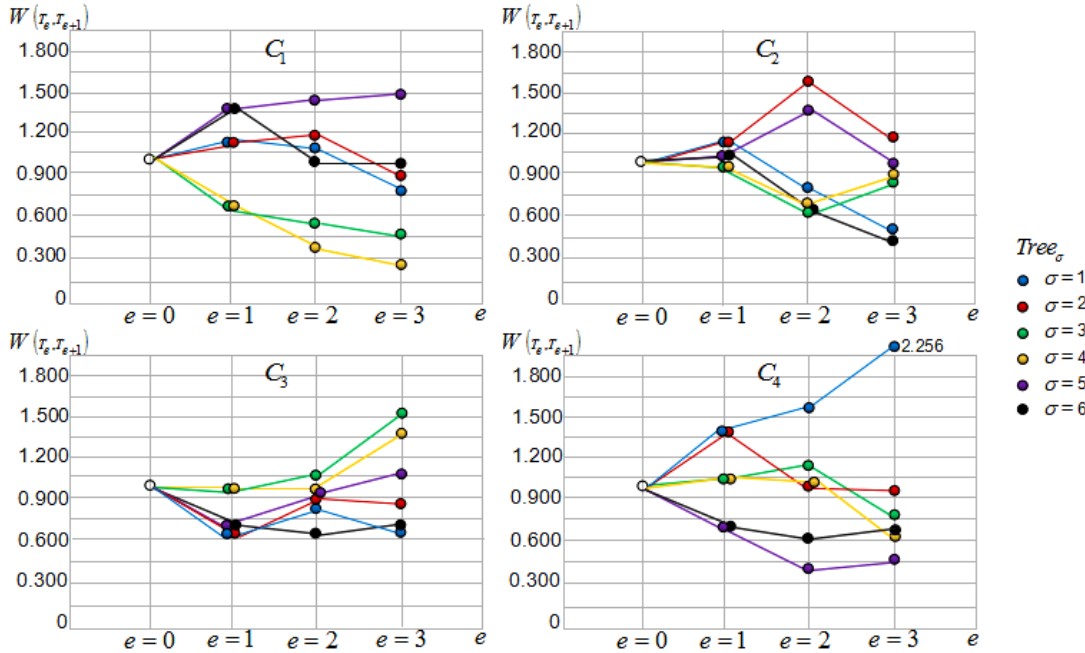

**Figure 9.** Each tourist sight sub-interval motive iteration value tendency curves.

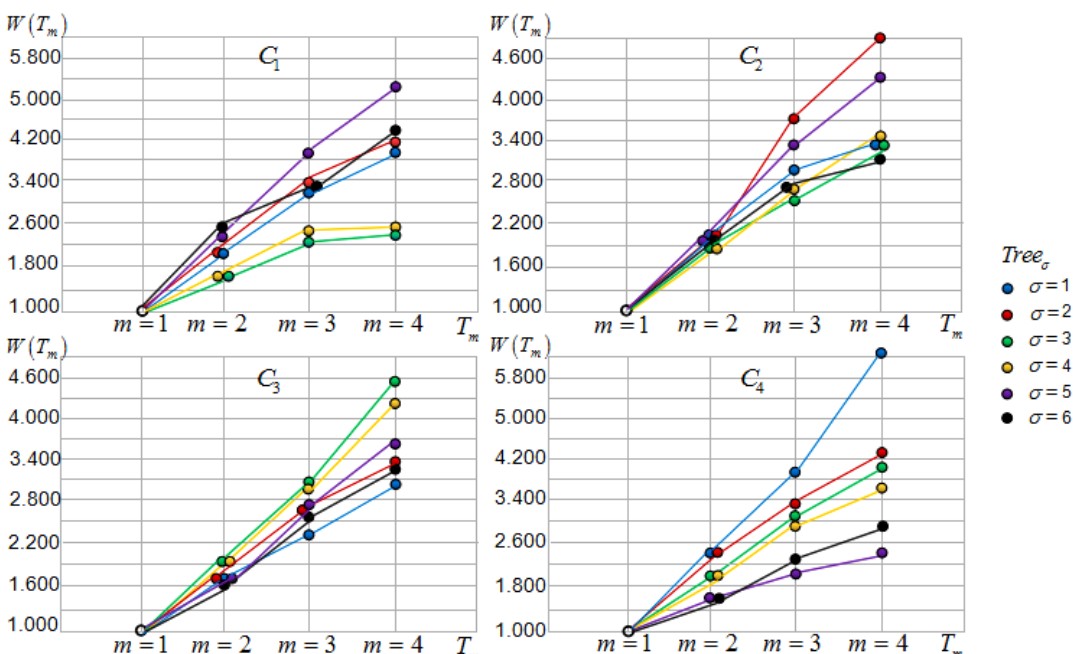

**Figure 10.** Each tourist sight interval motive iteration value and function curves.

### 4.2.2. Data Result Analysis and Discussion

Motive Iteration Cluster $C_v$ and Child Node Cluster $C\left(\neg T_e\right)$

As to Figure 7, the selected four tourist sights are set as father nodes to form child node clusters $C\left(\neg T_e\right)$ and the motive iteration cluster $C_v$, respectively. Since the initial father nodes are different from each other, the child node clusters that are formed have different distribution shapes.

In terms of geographical distribution, child node clusters $C\left(\neg T_1\right)$ and $C\left(\neg T_2\right)$ formed by tourist sights $T_1$ and $T_2$ have a long zonal distribution with a large opening angle directed northwest and

southeast. The child node clusters $C(\neg T_3)$ and $C(\neg T_4)$ formed by $T_3$ and $T_4$ have a comparatively short zonal distribution with a small opening angle directed southeast and northwest. Regarding the different shapes, opening angles and directions, the zonal distributions of $C(\neg T_1)$ and $C(\neg T_2)$ that account for child nodes $T_2$, $T_3$, $T_4$ and $T_1$, $T_3$, $T_4$ are relatively discrete in spatial clustering distribution. Child nodes $T_1$, $T_2$, $T_4$ and $T_1$, $T_2$, $T_3$ are relatively concentrated in spatial clustering distribution. In general, a more concentrated child cluster is much more beneficial because they have shorter times and ferry distances between tourist sights within the cluster, more convenient ferry access, lower taxi costs, and less traffic congestion, all of which contribute to generating a globally optimal solution. Regarding the iterating result, this simulation experiment's global optimal solution appears in motive iteration cluster $C_4$ formed by the tourist sight $T_4$ father node.

Motive Iteration Factor $c$ and Disturbance Factor $\varepsilon$

Considering Table 4 and Figure 8, the motive iteration factor $c$ and disturbance factor $\varepsilon$ vary in different sub-intervals. In the first group, sub-interval $H(T_1,T_4)$ has the highest factor $c$ value while sub-interval $H(T_3,T_4)$ has the lowest factor $c$ value. Sub-interval $H(T_2,T_4)$ has the lowest $-\varepsilon$ value, which has the strongest disturbance influence. Sub-interval $H(T_1,T_2)$ has the highest $-\varepsilon$ value with the weakest disturbance influence. In terms of the factor value, according to the clustering principle, sub-intervals $H(T_1,T_3)$, $H(T_2,T_3)$, $H(T_2,T_4)$ and $H(T_3,T_4)$ are clustered in one group, while $H(T_1,T_2)$ and $H(T_1,T_4)$ are clustered in another group.

In the second group, all sub-intervals factor $c$ values are 0.400 except that of $H(T_1,T_3)$, which is 0.300. Sub-interval $H(T_2,T_3)$ has the lowest $-\varepsilon$ value, whose disturbance influence is the strongest. Sub-intervals $H(T_1,T_2)$ and $H(T_1,T_4)$ have the highest $-\varepsilon$ value, whose disturbance influence is the weakest. Sub-intervals $H(T_1,T_3)$, $H(T_2,T_3)$ and $H(T_3,T_4)$ are clustered in one group, whereas sub-intervals $H(T_1,T_2)$, $H(T_1,T_4)$ and $H(T_2,T_4)$ are clustered in another group.

In the third group, all sub-intervals factor $c$ values are 0.100. Sub-interval $H(T_3,T_4)$ has the lowest $-\varepsilon$ value, whose disturbance influence is the strongest. Sub-interval $H(T_1,T_2)$ has the highest $-\varepsilon$ value, whose disturbance influence is the weakest. Sub-intervals $H(T_1,T_3)$, $H(T_1,T_4)$, $H(T_2,T_3)$ and $H(T_2,T_4)$ are clustered in one group. The other two sub-intervals form two clusters.

In the fourth group, sub-interval $H(T_1,T_4)$ has the lowest factor $c$ value, whereas sub-interval $H(T_1,T_3)$ has the lowest factor $c$ value. Sub-intervals $H(T_1,T_2)$ and $H(T_1,T_4)$ have the highest $-\varepsilon$ value, whose disturbance influence is the weakest. Sub-interval $H(T_3,T_4)$ has the lowest $-\varepsilon$ value, whose disturbance influence is the strongest. Sub-intervals $H(T_1,T_2)$ and $H(T_1,T_4)$ are clustered in one group, whereas sub-intervals $H(T_1,T_3)$, $H(T_2,T_3)$, $H(T_2,T_4)$ and $H(T_3,T_4)$ are clustered in another group.

In the fifth group, sub-interval $H(T_1,T_4)$ has the highest factor $c$ value, whereas sub-interval $H(T_3,T_4)$ has the lowest factor $c$ value. Sub-intervals $H(T_1,T_2)$, $H(T_1,T_4)$, and $H(T_2,T_4)$ have the

highest $-\varepsilon$ value, whose disturbance influence is the weakest, and they are clustered in one group. Sub-intervals $H\left(T_1,T_3\right)$, $H\left(T_2,T_3\right)$, and $H\left(T_3,T_4\right)$ have the lowest $-\varepsilon$ value, whose disturbance influence is the strongest, and they are clustered in one group.

Decision Tree Sub-Interval Motive Iteration Value $W\left(T_e,T_{e+1}\right)$

As can be observed from Table 5 and Figure 9, the motive iteration clusters generated from different tourist sight father nodes have large differences in terms of the sub-interval motive iteration values, the $W\left(T_e,T_{e+1}\right)$ output values, and the tendency curves. Each sub-interval's output value is determined by the previous sub-interval, and the value fluctuates. The values randomly vary up and down with the change in tourists' travel times and locations, which are determined by the sub-intervals initial value and factors $c$ and $\varepsilon$.

Compare the four motive iteration cluster $W\left(T_e,T_{e+1}\right)$ values and the tendency curves of the cluster $C_3$ generated from tourist sight father node $T_3$. These are most concentrated, which accounts for why the influence of each cluster $C_3$ decision tree relative tour routes on tourists is similar and in the same level. Tourists can choose any one of the tour routes and obtain the same motive benefit satisfaction.

Motive iteration cluster $W\left(T_e,T_{e+1}\right)$ tendency curves for clusters $C_1$, $C_2$ and $C_4$ are relatively discrete. For cluster $C_1$, the tendency curves of the decision trees $\sigma=1,2,6$ relative tour routes are close and can be clustered in one group, and the tendency curves of the decision trees $\sigma=3,4$ relative tour routes are close and can be clustered in another group. The tendency curves of the decision tree $\sigma=5$ relative tour route can be clustered in one group, and it has the greatest influence on tourists' motive benefit satisfaction. Regarding cluster $C_2$, the tendency curves of the decision trees $\sigma=2,5$ relative tour routes are close and can be clustered in one group, and the tendency curves of the decision trees $\sigma=1,3,4,6$ relative tour routes are close and can be clustered in another group. Regarding cluster $C_4$, the tendency curves of the decision trees $\sigma=2,3,4$ relative tour routes are close and can be clustered in one group, and the tendency curves of the decision trees $\sigma=5,6$ relative tour routes are close and can be clustered in another group. The tendency curves of the decision tree $\sigma=1$ relative tour route can be clustered in one group, and it has the greatest influence on tourists' motive benefit satisfaction. The tour routes clustered in one group have a similar influence on tourists' motive benefit satisfaction. Tourists may choose any one of the tour routes and obtain the same motive benefit satisfaction.

Decision Tree Interval Motive Iteration Value $W\left(T_m\right)$

Considering Table 5 and Figure 10, the motive iteration clusters generated from different tourist sight father nodes have large differences in the interval motive iteration values. In terms of output value $W\left(T_m\right)$, each motive iteration cluster's function $W\left(T_m\right)$ is monotonically increasing. Comparing four groups of motive iteration $W\left(T_m\right)$ output values, the tendency curves of the cluster $C_3$ generated from tourist sight father node $T_3$ are the most concentrated, because the cluster sub-intervals' tendency curves are the most concentrated. It also accounts for why the influence of each cluster $C_3$ decision tree relative tour routes for tourists is similar and in the same level. Tourists may choose any one of the tour routes and obtain the same motive benefit satisfaction. Regarding cluster $C_1$, the tendency curves of the decision trees $\sigma=1,2,6$ relative tour routes are close and can be clustered in one group, and the tendency curves of the decision trees $\sigma=3,4$ relative tour routes

are close and can be clustered in another group. The tendency curves of the decision tree $\sigma=5$ relative tour route can be clustered in one group, and it has the greatest influence on tourists' motive benefit satisfaction. Regarding cluster $C_2$, the tendency curves of the decision trees $\sigma=2,5$ relative tour routes are close and can be clustered in one group, and the tendency curves of the decision trees $\sigma=1,3,4,6$ relative tour routes are close and can be clustered in another group. Regarding cluster $C_4$, the tendency curves of the decision trees $\sigma=2,3,4$ relative tour routes are close and can be clustered in one group, and the tendency curves of the decision trees $\sigma=5,6$ relative tour routes are close and can be clustered in one group. The tendency curves of the decision tree $\sigma=1$ relative tour route can be clustered in one group, and it has the greatest influence on tourists' motive benefit satisfaction. The tour routes clustered in one group have a similar influence on tourists' motive benefit satisfaction, and tourists may choose any one of the tour routes and obtain the same motive benefit satisfaction.

The Motive Iteration Cluster Local Optimal Solution $W[c_v]$ and Global Optimal Solution $W[c_v]^{\max}$

Table 5 presents the motive iteration cluster descending sub-vector $\vec{K}_a$ and motive iteration cluster descending vector $\vec{R}_b$ that are obtained.

- As to cluster $C_1$, $\vec{K}_1 = [5.322 \quad 4.362 \quad 4.145 \quad 4.052 \quad 2.700 \quad 2.323]$;
- As to cluster $C_2$, $\vec{K}_2 = [4.904 \quad 4.362 \quad 3.485 \quad 3.394 \quad 3.393 \quad 3.103]$;
- As to cluster $C_3$, $\vec{K}_3 = [4.567 \quad 4.243 \quad 3.633 \quad 3.347 \quad 3.041 \quad 3.017]$;
- As to cluster $C_4$, $\vec{K}_4 = [6.221 \quad 4.260 \quad 3.923 \quad 3.605 \quad 2.955 \quad 2.498]$;
- Cluster $C_1 \sim C_4$, $\vec{R} = [6.221 \quad 5.322 \quad 4.904 \quad 4.567]$.

In Figure 10, the six best values of all four clusters are shown. The highest one is the cluster's local optimal solution $W[c_v]$. In the four motive iteration clusters, the decision trees $\sigma=5$, $\sigma=2$, $\sigma=3$ and $\sigma=1$ motive iteration values are local optimal solutions $W[c_v]$; they are $W[c_1]=5.322$, $W[c_2]=4.904$, $W[c_3]=4.567$ and $W[c_4]=6.221$, respectively. According to the definition, motive iteration cluster global optimal solution $W[c_v]^{\max}$ is $\max W[c_v]$, it is thus $W[c_4]=6.221$ whose relative tour route is the first decision tree in cluster $C_4$, shown as the red route in Figure 11.

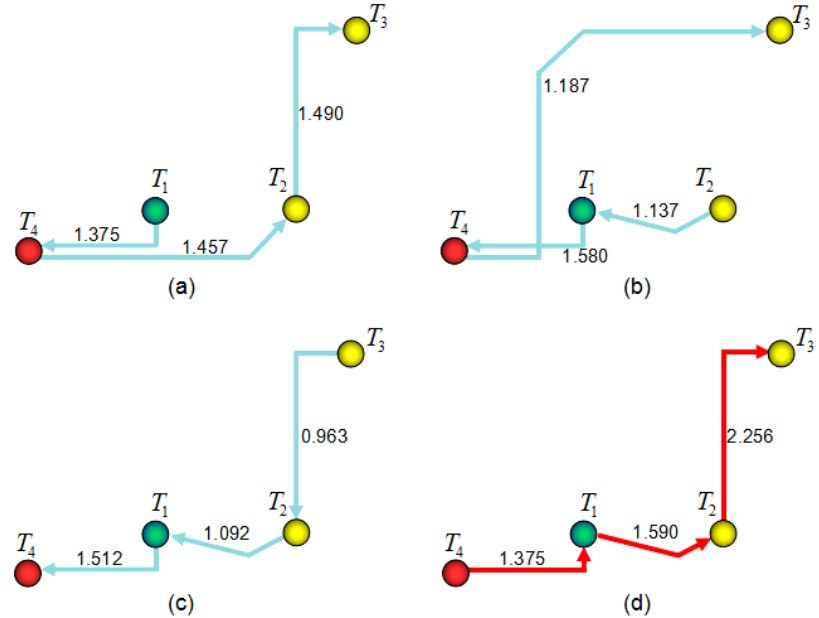

**Figure 11.** Tour routes for motive iteration decision tree local optimal solutions.

The local optimal solution $W[c_v]$ is the iteration value of the cluster optimal route. If tourists choose one starting tourist sight for the trip, the smart machine will output all the tour routes related to the cluster as descending sub-vector $\vec{K}_a$ elements and highly recommends the first element tour route for tourists. Considering a tourist's own needs and interests, they choose the tour route by themselves. For example, a tourist obtains accommodation next to $T_2$ the Erqi memorial and he takes this tourist sight as the starting one to visit. In this case, since the tour route is $T_2 - T_1, T_4, T_3$, which is Erqi memorial, Bishagang park, Zhongyuan Wanda, and Henan museum, and is the cluster's optimal tour route, the smart machine will highly recommend it to the tourist. If tourists do not choose the starting tourist sight, the smart machine will output all the tour routes related to the cluster descending vector $\vec{R}_b$ elements and specifically recommends the first element tour route for tourists, considering the tourist's own needs and interests, and they choose a tour route themselves. In the simulation experiment, the smart machine highly recommends the cluster $C_4$ tour route $T_4 - T_1, T_2, T_3$ which is Zhongyuan Wanda, Bishagang park, Erqi memorial, and Henan museum, the red route. Figure 11 shows the relative tour routes of the motive iteration decision tree local optimal solutions, of which the red noted route is the global optimal solution tour route. The decision tree father nodes for Figure 11a–d are tourist sights $T_1 \sim T_4$, respectively.

The Advantage of the Algorithm

As to the study, the algorithm designed in the paper could plan optimal tour routes for tourists and help them to get best motive benefits. Firstly, it considers tourists' individualized needs and interests. The smart machine designed in the study can automatically select interested tourist sights and plan optimal routes according to the basic information provided by tourists, which is convenient and intelligent. The planned routes combine factors and disturb factors of GIS service, which are all genuine and precise factors tourists must consider and deal with during the whole trip. In this aspect, tourists will not plan routes by themselves; rather, all the optimal routes are planned by the smart machine. This process is better than the procedure whereby tourists find mass information on the Internet and plan routes by themselves, in which many key factors may be neglected and cannot provide the best travel experience and motive benefits. Meanwhile, the smart machine only considers tourists' profits and interests, but travel companies mainly consider their own profits to earn more

money, which will neglect tourists' needs and interests. In all, the optimal routes of smart machine can meet the needs and interests of tourists and thus provide better information than mass tourism data on the Internet and travel companies.

## 5. Conclusions

Some critical issues and existing problems in smart tourism and tourism GIS are discussed and analyzed in the study. Mining the most valuable tour route knowledge from big data-level information is the key to increasing the motive benefit satisfaction. When planning a trip and tour routes, tourists are usually unfamiliar with a strange city and its tourism services information. Travel agencies provide planned tour routes for tourists to gain profits, and they insufficiently consider tourists' needs and interests since they provide group tours with fixed schedules, offer limited ferry transportation ways, and confine the range of activities for tourists.

Referring to Reference [5]'s concept of using a scalable geospatial analysis based on cloud computing platform to detect tourism destinations, this study establishes basic city tourist sight data information and GIS data as independent variables to build a feature interest tourist sight extracting matrix, which is not scalable. Reference [5] uses a cloud computing platform, whereas this study develops a new calculating system to obtain iteration values.

Referring to Reference [17]'s concept of oriented spanning trees, this study also generates spanning trees. Reference [17] adds genetic and multi-criteria thoughts to solve the path problems; in this study, there is one criterion, which aims to determine the maximum iteration value.

Representative tourist sights and service functions are integrated and are used as a data resource to build the algorithm. The tourist sight classification is a subset, and it is an effective means to group and segment tourists' needs and interests. It is also the basis upon which to build the tourist sight spatial interest field mapping model.

An age index is used to group tourists because this standard has broad coverage and strong representation because similarly aged people have similar interests. The developed smart tourist sight extracting algorithm model is highly random and a strict logic is used in the algorithm, covering all the tourist sights, with each tourist sight having the same probability of being selected. Considering one-day trips, in order to ensure that tourists have an enjoyable trip experience at tourist sights to obtain the best motive benefit satisfaction, the smart machine sets an upper limit for the number of selected tourist sights and then stores, manages, and plans tourist sights and tour routes accordingly.

Referring to Reference [18]'s approach to solving shortest-distance problems, in designing the algorithm and smart machine, this study supplements more details to meet a majority of the tourists' needs. Two principles are applied; one is the principle of proximity, and the other is the principle of completely random. Considering these two principles, a smart motive iteration decision tree algorithm is designed and developed. A quantitative method is used to evaluate the motive iteration trees and tour routes generated from different tourist sight father nodes and the results are used as the basis for smart machine recommendations for tourist sights and tour routes.

Compared with Reference [39]'s application of spatial partitioning and k-means clustering, the concept of clustering is also used in this study. Reference [39] applies k-means clustering to habitat occupation in Propithecus perrieri. Similarly, the m-central point clusters are developed, where each tourist sight is used as father node to generate decision trees. When a tourist chooses situation one and starts at the closest tourist sight, only one cluster will be studied and used to determine the optimal tour route, which will decrease the cost. When a tourist chooses situation two and randomly chooses a starting tourist sight, the smart machine needs to determine the global optimal solution and recommend the best tour route for tourists.

The recommended tour routes obey the optimum principle, and individualized interests and needs are considered. In tour route planning, it avoids tourists' subjective cognition and considers mainly individualized needs most, in addition to objective conditions. The methodology does not seek to pursuit profit as do travel agencies; instead, it is based on serving tourists. Regarding massive and big data-level tourism information, this study presents a method to access valuable and

concealed tour route knowledge, which are relevant to tourists' needs and interest. The algorithm developed in the study is practical and its performance are an effective examination of data mining in mass tourism data.

**Author Contributions:** This research was completed jointly by all the authors. D.Z. collected and processed the original data. Y.Z. and G.F. mainly conducted and designed the experiments. X.Z. performed the experiments and wrote the paper. S.L. assisted with the experimental design. X.Z. and Y.Z. conducted the data collection and analysis.

**Funding:** This research was funded by the National Natural Science Foundation of China (Grant No. 41571399, 41501446, 41771487 and 41704006), Training Program for Young Backbone Teachers of Henan Higher Education Institute (Grant No. 2017GGJS196), Henan Scientific and Technological Project (Grant No. 182102210554) and Open-end Fund of State Key Laboratory of Geo-Information Engineering (Grant No. SKLGIE2018-ZZ-6)

**Conflicts of Interest:** The authors declare no conflicts of interest.

## Abbreviations

| | Meaning | Meaning of Subscript and Superscript |
|---|---|---|
| $P$ | tourist sight spatial data set | None |
| $P_r$ | tourist sight spatial data subset | $r$ : subset number |
| $P_r Q_s$ | Subset element tourist sight | $s$ : element tourist sight number in subset |
| $t$ | tourist sight classification number | None |
| $p_r$ | tourist sight number of the subset | $r$ : subset number |
| $\vec{P}_r$ | tourist sight extracting base vector | $r$ : subset number |
| $\mathbf{P}$ | tourist sight extracting matrix | None |
| $G_i$ | Age group classification | $i$ : age group number |
| $n$ | Number of people in statistics | None |
| $n_i$ | Age group $G_i$ number of people | $i$ : age group number |
| $age$ | Age index | None |
| $k_{i,r}$ | Number of people visiting $G_i$ | $i$ : age group number <br> $r$ : subset number |
| $\omega_{i \cdot r}$ | Visited rate of $P_r$ | $i$ : age group number <br> $r$ : subset number |
| $m$ | Number of tourist sight to be visited | None |
| $T$ | Selected tourist sight set | None |
| $T_e$ | Selected tourist sight set element | $e$ : Selected tourist sight set number |
| $W(T_e)$ | Motive iteration function | $e$ : Selected tourist sight set number |
| $H(T_e, T_{e+1})$ | Motive iteration sub-interval | $T_e$ :Selected tourist sight set element <br> $e$ : Selected tourist sight set number |
| $W(T_e, T_{e+1})$ | Sub-interval motive iteration value | $T_e$ :Selected tourist sight set element <br> $e$ : Selected tourist sight set number |
| $H(T_1, T_m)$ | Motive iteration interval | $T_m$ : The final tourist sight to be visited |
| $W(T_1, T_m)$ | Interval motive iteration value | $T_m$ : The final tourist sight to be visited |

| | | |
|---|---|---|
| $c$ | Motive iteration factor | None |
| $\delta$ | Specific factor | None |
| $\varepsilon$ | Motive iteration disturbance factor | None |
| $\mu_1$ | Specific disturbance factor | None |
| $Tree_\sigma$ | Motive iteration decision tree | $\sigma$ : Tree number |
| $C\left(_{\neg T_e}\right)$ | Child node cluster | $T_e$ :Selected tourist sight set element <br> $e$ : Selected tourist sight set number |
| $C_v$ | Motive iteration cluster | $v$ : Cluster number |
| $W\left[c_v\right]$ | Decision tree local optimal solution | $C_v$ :Motive iteration cluster |
| $W\left[c_v\right]^{max}$ | Decision tree global optimal solution | $C_v$ :Motive iteration cluster |
| $\vec{K}_a$ | Motive iteration cluster descending sub-vector | $a$ : Motive iteration cluster descending sub-vector number |
| $\vec{R}_b$ | Motive iteration cluster descending vector | $b$ : Motive iteration cluster descending vector number |
| $\forall$ | arbitrary | None |
| $\cup$ | Join | None |
| $Z^+$ | Positive integer | None |
| $R^+$ | Positive real number | None |

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
