# Peer review of "Individualized Tour Route Plan Algorithm Based on Tourist Sight Spatial Interest Field"

_ijgi, doi:10.3390/ijgi8040192_

Round 1
Reviewer 1 Report
First of all, I appreciate to the authors for making efforts to carry out the changes by the referees. I think the authors did a good job in clarifying the queries that this manuscript is substantially improved.
Author Response
Dear reviewer,
Thank you for your suggestions. We will continue working hard and make good efforts.
Yours truly, all authors
Reviewer 2 Report
It seems that authors have addressed reviewers' comments very well. However, the importance of spatial approach in tourism and community planning should be emphasized based on relevant literatures in recreation, park and tourism. Please see several key references below.
Kim, J., Thapa, B., & Jang, S. (2019). GPS-based mobile exercise application: An alternative tool to assess spatio-temporal patterns of visitors' activities in a National Park. Journal of Park and Recreation Administration, 37(1), 124-134.
kang, S., Lee, G., Kim, J., & Park, D. (2018). Identifying the spatial structure of tourism attraction system in South Korea using GIS and network analysis: An application of anchor-point theory. Journal of Destination Marketing & Management, 9, 358-370.
Kim, J., & Nicholls, S. (2016). Influence of the measurement of distance on assessment of recreation access. Leisure Sciences, 38(2), 118-139.
Kang, S., Kim, J., & Nicholls, S. (2014). National tourism policy and spatial patterns of domestic tourism in South Korea. Journal of Travel Research, 53(6), 791-804.
Author Response
Dear reviewer,
Thank you for your advice. We have found the references you mentioned and downloaded them all. All the authors did an earnest study on all the references and learned a lot. In the paper, we found out all the discussion and experiment that should be emphasized and noted the references as [56][57][58][59] respectively. Thank you for your help and we will continue making great efforts to make our work better.
Yours turly, all authors.
This manuscript is a resubmission of an earlier submission. The following is a list of the peer review reports and author responses from that submission.
Round 1
Reviewer 1 Report
This manuscript, “Individualized tour guide route motive iteration decision tree algorithm based on sight spot space interest field” developed smart tour guide route motive iteration decision tree algorithm to meet tourists’ needs and interests. Overall, the topic and purpose of the study are interesting, but more references and detailed reasons are needed for each step of building the algorithm.
Abstract
Page 1, line 24: change Information to information
Introduction
Overall, literature and references should be strengthened, as so many topics are dealt with (smart tourism, tourist experience, GIS, tourist motive benefit, etc.).
Page 1, lines 38-40 (The target of smart tourism is…): Add citations
Pages 1-2, lines 42-44 (It takes advantage of cloud…): In one sentence, explain in detail what is the most concerned tourism information on the Internet.
Page 2, line 66 (The technique developing orientation…): Add citations.
Page 2, line 77 (The mode is usually…):Add citations
Page 2, lines 90-91 (According to age index): Explain in detail why this study simply divides tourists into three groups. This study was to solve the problems in which previous ways that provide tour routes have not met individualized needs and interest (lines 85-87). Unlike the research background and purpose that individualized service is needed, tourists are largely divided into three groups, which conflict with the background and purpose of the study.
Methods
Page 5, lines 185-186 (Step 1 Tourist provide age index…): provide supportive reasons why tourists should choose the number of sight spots first, despite the number of sight spots tourists can visit during a day may vary according to the characteristics of destination. Furthermore, authors should describe why the number of selected sight spot cannot exceed 5 and add citations.
Page 6, lines 220-222 (In this process, ferry…): Add citations for each element (ferry distance, road traffic condition, and the convenience of GIS service tourists’ subjective psychology) because this study applied these elements to measure motive iteration factor c.
Page 7, line 267 (the fourth bullet point: average waiting time..): Describe average waiting time for what.
Page 13, line 408: Explain the reason why this study is suitable for city urban tour
Results and discussion
It is necessary to verify that the optimal path of smart machine can meet the needs and interests of tourists and thus provide better information than travel companies or mass tourism data on the Internet.
Conclusions
Page 22, lines 631-632 (Travel agencies provide…): Explain the reasons why travel agencies’ plans are hardly considers tourist’ needs and interests. Without proper reasons, it is hard to say that a travel agency cannot meet the needs and interests of tourists due to the company's pursuit of profits.
Author Response
Revision report
Dear expert,
Thank you for your great help on reviewing the paper and providing the valuable points for all the authors. We have seriously read and analyzed your review and comments you mentioned. After discussing, now we answer and explain the questions from point by point in the following revision report. Thank you.
1. Abstract
Answer: The word”Information” has been modified into”information”in the paper.
2. Introduction
(1) Page 1, lines 38-40 (The target of smart tourism is…): Add citations
Answer: Add following citations for line 38-40 and note them in paper.
[3] Balmford, A.; Beresford, J.; Green, J.; Naidoo, R.; Walpole, M.; Manica, A. A global perspective on trends in nature-based tourism. PLoS Biol. 2009, 7, e1000144.
[4] Page, S.J. Transport and Tourism: Global Perspectives; Pearson: Harlow, UK, 2005.
(2) Pages 1-2, lines 42-44 (It takes advantage of cloud…): In one sentence, explain in detail what is the most concerned tourism information on the Internet.
Answer: Add following one sentence after line 44 to explain the most important information.
Of all the information, sight spot classification, locations, distinguishing features, transportation service around, traffic and accommodation fees are the most concerned ones.
(3) Page 2, line 66 (The technique developing orientation…): Add citations
Answer: Add following citations for line 66 and note them in paper.
[5] Zhou, X.; Xu, C.; Kimmons, B. Detecting tourism destinations using scalable geospatial analysis based on cloud computing platform. Comput. Environ. Urban Syst. 2015, 54, 144–153.
[6] Cheng, T.; Haworth, J.; Wang, J. Spatio-temporal autocorrelation of road network data. J. Geogr. Syst. 2012, 14, 389–413.
(4) Page 2, line 77 (The mode is usually…):Add citations
Answer: Add following citation for line 77 and note it in paper.
[7] Kruger, M.; Viljoen, A.; Saayman, M. Who Visits the Kruger National Park, and Why? Identifying Target Markets. J. Travel Tour. Mark. 2017, 34, 312–340.
(5) Page 2, lines 90-91 (According to age index): Explain in detail why this study simply divides tourists into three groups. This study was to solve the problems in which previous ways that provide tour routes have not met individualized needs and interest (lines 85-87). Unlike the research background and purpose that individualized service is needed, tourists are largely divided into three groups, which conflict with the background and purpose of the study.
Answer: As to point (5), here are illustrations.
1) As to various sorts of sight spots, according to statistics big data, different tourist groups have discrepant interest tendency, and the same group have the similar one. We choose age index as the main factor to divide tourists because this is the most critical step to set up sight spot spatial interest field mapping model. If tourists are divided into different groups by two or more indexes, the mapping model will be disordered and the interest tendency will not be clear and apparent.
2) The individualized needs and interests mentioned in the paper mainly include sight spot number, classification, specific sight spot selected, tour route and traffic service. In designing the algorithm, we combine the factors together and try to meet individual needs. Therefore, the factors studied in the paper cannot be too abundant, which will enhance the difficulty in designing the algorithm and developing smart machine since one study of smart machine design cannot meet all needs of each single tourist, as there are tens of thousands of tourists and needs. Thus, the designed algorithm aims to meet individualized needs to the greatest extent. In future study, we will do further more research.
3) The smart machine algorithm designed in the paper includes two modes in selecting interested sight spots for tourists to visit. We try to deal with the two modes of situations to meet the individualized needs and interests.
The first mode, tourists absolutely believe in and depend on smart machine, in order to get the most convenient service, they do not select sight spots by themselves. So they only provide the number and personal information to the smart machine. The machine will automatically select interested sight spots for tourists. The most important index that the machine refers to is the age index, because different age groups have discrepant interests while the same age group have the similar ones according to the statistics big data. This study mainly does experiment on age index, which is the most important one or the most commonly used one. If the study concerns more indexes, there will be no standard and will make confusion.
The second mode, tourists believe in smart machine but they also want to take part. They can either select sight spots one by one all by themselves, or they can modify the selecting results of smart machine to exchange certain ones. If tourists do not like certain sight spot, they can tell the machine and change it directly. In this case, it sufficiently considers individualized needs.
In all, we try to meet individualized interests and needs in the algorithm design and smart machine developing, however, any algorithm and machine can only consider overwhelming majority of tourists’ needs, may be a few part of tourists will not be satisfied with the results provided, as there is no 100% satisfaction. In the further study, we will continue working on the issue and make it a better one.
3. Methods
(1) Page 5, lines 185-186 (Step 1 Tourist provide age index…): provide supportive reasons① why tourists should choose the number of sight spots first, despite the number of sight spots tourists can visit during a day may vary② according to the characteristics of destination. Furthermore, authors should describe③ why the number of selected sight spot cannot exceed 5 and add citations④.
Answer: As to point (1), here are illustrations.
① Actually, choosing the number of sight spots first is one of the programming designs for the smart machine, the aim of the thought is to make the tour route plan more convenient and intelligent, tourists only need to input the number, the smart machine will immediately plan an optimal route for the tourist, and this is also the core thought and aim of the study. As to the two modes mentioned above, when tourists choose mode 1, after providing basic personal information such as gender, age, etc., tourists should only provide the number of sight spots. The smart machine will automatically select sight spots according to the algorithm designed in the study, here we add the thought and procedure of machine learning.
② Because of the procedure of machine learning, if tourists choose mode 2, after the smart machine selecting the sight spots, tourists can change certain sight spots if he is not interested in them because each person’s needs and interests as well as the number of sight spots he wants to visit are various. This is also the thought of the design for programming of the smart machine.
③ In designing the algorithm for smart machine in the study, the urban tour’s traveling time is one day, as we set. Within one day, in order to have the best travel experience and meanwhile avoid fatigue to get the best motive benefits, we set the maximum number of 5 for selected sight spots, including the morning, afternoon and night. Usually, tourists may firstly have a good sleep and rest before travel, and they may plan two sight spots in the morning, two in the afternoon and one at night, or only in daytime without night. If the number exceeds 5, time will be very tight, and the time spend on each sight spot will be very short and insufficient, which will influence travel experience.
④ We’ve added citations [10][18][24] in the paper.
(2) Page 6, lines 220-222 (In this process, ferry…): Add citations for each element (ferry distance, road traffic condition, and the convenience of GIS service tourists’ subjective psychology) because this study applied these elements to measure motive iteration factor c.
Answer: Add following citations for elements.
[52] Hall, C.M.; Le-Klahn, D.T.; Ram, Y. Tourism, Public Transport and Sustainable Mobility; Channel View
Publications: Bristol, UK, 2017.
[53]Cheng, Y.H.; Chen, S.Y. Perceived accessibility, mobility, and connectivity of public transportation systems. Transp. Res. Part A Policy Pract. 2015, 77, 386–403.
[54]Thill, J.C. Geographic information systems for transportation in perspective. Transp. Res. Part C Emerg. Technol. 2000, 8, 3–12.
[55]Benenson, I.; Martens, K.; Rofé, Y.; Kwartler, A. Public transport versus private car GIS-based estimation of accessibility applied to the Tel Aviv metropolitan area. Ann. Reg. Sci. 2011, 47, 499–515.
(3) Page 7, line 267 (the fourth bullet point: average waiting time..): Describe average waiting time for what.
Answer: Here is the explanation for the average waiting time.
In many cities, when you take a taxi, you should pay two parts of fees. One part is charged by the total distance you travel, the other part is the average waiting time. During the service, the taxi may meet traffic jam or red traffic light and then the taxi will stop to wait the jam or light, so the average waiting time is the quotient of total waiting time to taxi stop times. This explain is added in the paper.
(4) Page 13, line 408: Explain the reason why this study is suitable for city urban tour
Answer: Here is the explanation and add it in the paper.
Urban tour is one of the hot spots of smart tourism. The aim of the study is to find out optimal tour routes for tourists to get best motive benefits. So the study range is merely one certain city downtown, and all the factors and disturb factors come from urban GIS service and concept. The routes planned in the study are all based on city urban roads and avenues. The sight spots selected for the original data source are all urban sight spots which are located in the city but not in the outskirts. In the aspect of space concept, sight spots are close to each other, tourists can visit all the selected sight spots in one day. Secondly, in one city, the factors and disturb factors mentioned in the study are identical, it is precise to use the identical factors as parameters to do the study as different cities have different conditions. Second, this study is suitable for city urban tour but not suitable for rural tour, as some factors used in the study do not exist in country such as taxi, traffic lights, subway, urban bus, etc. Meanwhile, it is also not suitable for cross-region or cross-city tour, because the distance is too long, and the travel time will be too long either, as to this reason, the number of cross-region or cross-city tour’s selected sight spots will be less than city urban tour. Also, some factors such as urban taxi, subway, urban bus,etc.are not suitable for this type of tour.
4. Results and discussion
(1) It is necessary to verify that the optimal path of smart machine can meet the needs and interests of tourists and thus provide better information than travel companies or mass tourism data on the Internet.
Answer: Add the following verifying in the paper.
As to the study, the algorithm designed in the paper can plan optimal tour routes for tourists and help them to get best motive benefits. Firstly, it considers tourists’ individualized needs and interests. The smart machine designed in the study can automatically select interested sight spots and plan optimal routes according to the basic information provided by tourists, which is convenient and intelligent. The planned routes combine factors and disturb factors of GIS service, which are all genuine and precise factors tourists must consider and deal with during the whole trip. In this aspect, tourists will not plan routes by themselves, all the optimal routes are planned by machine. It is better than the procedure that tourists find mass information on the Internet and plan routes by themselves, in which many key factors may be neglected and finally cannot get the best travel experience and motive benefits. Meanwhile, the smart machine only considers tourists’ profits and interests, but travel companies mainly consider their own profits to earn more money, which will neglect tourists’ needs and interests. In all, the optimal routes of smart machine can meet the needs and interests of tourists and thus provide better information than mass tourism data on the Internet and travel companies.
5. Conclusions
(1) Page 22, lines 631-632 (Travel agencies provide…): Explain the reasons why travel agencies’ plans are hardly considers tourist’ needs and interests. Without proper reasons, it is hard to say that a travel agency cannot meet the needs and interests of tourists due to the company's pursuit of profits.
Answer:This suggestion the expert put up forward is a good reminder for us. Because we cannot simply say that travel agencies’ plans hardly consider tourists’ needs and interests, they do consider tourists’ interests. But to some extent, it is not sufficient. They mainly consider group tour and majority people’s needs, not individualized needs. So the expression we used in the paper is not proper. So we do not only add the reasons, but modify the expressions. For example, we change “hardly” to “insufficiently”.
Tourists’ best motive benefits mainly come from the best time schedule, route schedule, sight spots selected, ferry transportation convenience and proper money cost on maximum travel experience, etc. Of all the factors, travel agencies hardly consider all of them, some travel agencies even consider few of them.
As to time schedule, route schedule and sight spots selected, since travel agencies usually make group tour, they provide fixed time and route schedule for tourists to choose, and each schedule is related to certain cost. Under this condition, tourists can only choose fixed and limited sorts of tour routes. Maybe one tour route contains several interested sight spots, but in this scheduled route, there may be just one or two or more sight spots that tourists do not like at all, but as it is a group tour, the loath sight spot(s) cannot be changed or canceled despite that one or minority part of tourists do not like certain sight spots. This is very hidebound. If tourists take the tour, they will still visit the loath ones and pay the same money, mentally, they may be unhappy or at least not be so satisfied with the schedule.
As to ferry transportation, in one city, there are urban buses, taxis, subways, boats, shared-bikes, shared-cars etc, for tourists to choose. Of all the traveling ways, they can provide tourists with different experience. Tourists can choose low cost way to travel, or more convenient and flexible way to travel. But travel agencies only provide tourists with group bus to travel and confine their range of activity. Some places tourists want to have a look is not allowed as time, range of activity and transportation are confined.
As to proper money cost on maximum travel experience, in group tour, one special tour route is related to one price. Of all the money tourists pay for travel agencies, one part is worth for the trip cost, the other part will be earned by travel agencies. But actually, tourists can spend less money to get the same travel experience not relying on travel agencies.

Reviewer 2 Report
The author(s) of the manuscript submitted suggest a work about individualized tour guide route motive iteration decision tree algorithm based on sight spot space interest field. The manuscript is clearly written and easy to follow, and it is within the journal scope, therefore I will suggest its publication after minor revisions.
General comments:
I would like to suggest the author(s) to put more extensive introduction for the general journal readers. Please change the format of line 64 that is not properly formatted.
Please give the objectives and hypotheses of this study in the end of introduction.
Please insert a small paragraph between Section 2 and Section 2.1. and between Section 4.2 and Section 4.2.1.
Conclusions should be included in a more concisely way and compared with similar studies. The discussion should be improved. There needs to be more comparative analysis with other studies.
Figures are not clear enough so they are necessary to increase their quality and size for the readability for the general journal readers. Also, there are too many figures overlapped with text that can be considered as redundancy.
Please include the references I have suggested and also update more recent researches.
J.S. Jeong, L. García-Mourno, J. Hernández-Blanco, F.J. Jaraíz-Cabanillas, 2014. An operational method to supporting siting decisions for sustainable rural second home planning in ecotourism sites. Land Use Policy, 41, 550-560.
T. Rahayuningsih, E.K.S. Harini Muntasib, L. Budi Prasetyo, 2016. Nature Based Tourism Resources Assessment Using Geographic Information System (GIS): Case Study in Bogor. Procedia Environmental Sciences, 33, 365-375.
References in the text and references list are particularly not followed by the journal guidelines and please include DOI if they have.
Author Response
Revision report
Dear expert,
Thank you for your great help on reviewing the paper and providing the valuable points for all the authors. We have seriously read and analyzed your review and comments you mentioned. After discussing, now we answer and explain the questions from point by point in the following revision report. Thank you.
1. I would like to suggest the author(s) to put more extensive introduction for the general journal readers. Please change the format of line 64 that is not properly formatted.
Answer: As to this suggestion, we add the following extensive introduction for general journal readers, including the introduction of smart tourism, tour route and planning, artificial intelligence method used in smart tourism. In line 64, we modify the line spacing as it looks like a bit insufficient.
Smart tourism is also called intelligent tourism. It uses techniques of cloud computing, internet of things, etc., via internet or mobile internet in portable terminal, to perceive information on tourism resource, tourism economy, tourism activity and tourists,etc., and then releases relative information for tourists to know about and refer to. Thus, according to the information, tourists can arrange time schedule and plan the trip during vacation. The building and developing of smart tourism will reflect in tourism experience, tourism management, tourism service and tourism marketing.
Tour route plan is the method to make a tour schedule, which takes shortest time to get best travel experience. It uses traffic routes to cascade different sight spots or tourism cities. The method of tour route plan is critical in making a good schedule to get the best motive benefits. Tour routes have the common features of comprehensiveness and flexibility. They are productions integrated by tourism attraction, ferry transportation, accommodation, catering, amusement and shopping,etc., which relates to many departments. Sight spots selected and schedule arrangement can be changed according to tourists’ interests, thus tour route plan is flexible.
2. Please give the objectives and hypotheses of this study in the end of introduction.
Answer: In the end of the introduction, we add objectives and hypotheses as follows.
To get the best motive benefits for tourists is the core objective, and this objective is set up by quantitative algorithm model as dependent hypothesis variable to find out the optimal tour routes, which is iterated by several important independent variables. The independent variables are factors and disturb factors, including critical sight spot information and GIS service information. After ensuring sight spot spatial interest field mapping model and selecting interested sight spots, all the factors and disturb factors are quantified and altered according to different motive iteration clusters and trees. Via iterating and outputting motive iteration values, the relative tour routes are all obtained, among which the maximum iteration value route is the optimal one for tourists, and the sub-optimal ones will also be displayed for tourists to select as there are different situations and project suggestions.
3. Please insert a small paragraph between Section 2 and Section 2.1. and between Section 4.2 and Section 4.2.1.
Answer: Add the following paragraph between Section 2 and Section 2.1
Interested sight spots are the basic data source to make tour route plan and get motive iteration values. In smart machine, interested sight spots are selected automatically and the interest tendency is the key for the machine to learn and recognize tourists’ needs. Thus, feature interests sight spot extracting algorithm based on interest field mapping model is set up first.
Answer: Add the following paragraph between Section 4.2 and Section 4.2.1.
Based on the algorithm models set up and the data sampling above , simulation experiment is performed and data result analysis is obtained.
4. Conclusions should be included in a more concisely way and compared with similar studies. The discussion should be improved. There needs to be more comparative analysis with other studies.
Answer: In the original conclusion, we modify and delete some fussy expressions and make the conclusion more concise.
The following expressions are modified.
① Age index is used as index to group tourists because this standard has the features of broad coverage, strong representation and similar age people having similar interests, etc.
② In planning tour routes, smart machine considers two situations, one is principle of proximity, the other is principle of completely.
③ Considering the two situations, smart motive iteration decision tree algorithm is designed and developed.
④ Representative sight spots and service functions are integrated, which is used as data resource to build the algorithm.
The following expressions are deleted.
① Tourism data information is massive and discrete and not all the data information is valuable knowledge.
② Valuable tour route knowledge is concealed in big-data level tourism information. Tourists can hardly get direct and most valuable tour experience and best motive benefit satisfaction.
③ It sets different sight spots as father nodes to build m-central point motive iteration clusters and child node clusters, and take the cluster as independent unit to plan tour routes. Set sight spot data information and GIS service data as quantitative indexes to iterate and output motive iteration values, and then store the values into descending vectors.
④ The usage of data is equal and authentic.
⑤ When the statistics number of people is sufficient, it can stands for the interest tendency of the very age group to the utmost extent so that the built model could be comprehensive, representative and integrated.
⑥ It is precise and targeted in recommending sight spots as it can select tourists’ interested sight spots to the utmost extent.
Add comparative analysis.
As to critical algorithm and method founded in the paper, add comparative analysis in the conclusion, including the following ones.
① Referring to citation[5]’s thought of using scalable geospatial analysis based on cloud computing platform to detect tourism destinations, this study sets basic city sight spot data information and surrounding GIS data as independent variables to build feature interest sight spot extracting matrix, which is not scalable. Citation[5] uses cloud computing platform, while this study develops a new calculating system to get iteration values.
② Referring to citation[17]’s thought of oriented spanning tree, this study also generates spanning trees. Citation[17] adds genetic and multi-criteria thoughts to solve path problems, in this study, criteria is single, which aims to find out maximum iteration value.
③ Referring to citation[18]’s thought to solve shortest-distance problems, in designing the algorithm and smart machine, this study supplements more details to meet majority tourists’ needs, in which, it considers two situations, one is principle of proximity, the other is principle of completely. Considering the two situations, smart motive iteration decision tree algorithm is designed and developed. Quantitative method is used to evaluate motive iteration trees and tour routes generated from different sight spot father nodes and the results are used as the standard for smart machine to recommend sight spots and plan tour routes.
④ Compared with citation[39]’s applying spatial partitioning and k-means clustering, the concept of clustering is also used in this study. Citation[39] uses k-means clustering to habitat occupation in Propithecus perrieri. Similarly, m-central point clusters are set up, each sight spot is used as father node to generate trees, and when tourists choosing situation one, only one cluster will be studied and used to find out the optimal tour route, which will save the cost. When tourists choosing situation two, smart machine needs to find out global optimal solution and recommend the best tour route for tourists.
5. Figures are not clear enough so they are necessary to increase their quality and size for the readability for the general journal readers. Also, there are too many figures overlapped with text that can be considered as redundancy.
Answer: According to expert’s suggestion, we find out that some figures are indeed unclear, which should be modified. We use software to increase figures’ quality and size. Meanwhile, original figure 5 is overlapped with text in which the tour route direction is also shown in figure 11(the last figure), and also the vectors’ figures are redundant.
Special explanation: Grid lines of Figure 8~figure 10 use light gray color in order to extrude color points. Absolute black grid lines are not proper because the color points are relatively small, while there are many connecting curves, black color will make visual confusion as we have tried black grid lines.
As to other figures, here we give following explanations.
Figure 1: It is used to show the mapping relationship between age groups and sight spot subsets. And each mapping relationship relates to one histogram. As the relationship is the mode of one-to-many, this figure is visualized and can help readers to understand the content.
Figure 2: The purpose of figure 2 is to show two definitions, they are motive iteration interval and sub-interval value. As the two definitions contain two levels and are relatively complex, readers may not understand the definitions only via words. So we add the figure here.
Figure 3: The purpose of figure 3 is to show that the motive iteration function is a discrete and monotonic increasing function, and its function values are discontinuous.
Figure 4: The purpose of figure 4 is to display different clusters, father nodes, child nodes and sight spots. It is visualized to tell readers the sight spot distribution, cluster shape, direction, etc. Only by reading words, readers may not know clearly all the information.
Figure 5: The purpose of figure 5 is to display different subset sight spots’ geospatial distribution and tell readers where the sight spots are. They can be found in electronic map.
Figure 6: Figure 6 is the built interest field mapping model. It is visualized and can be easily understood by readers. It is a supplement to table 3 and also an instantiation of figure 1’s mapping figure.
Figure 7: Figure 7 is an instantiation of figure 4 as it is used to show the result of simulation experiment.
Figure 8~Figure 11: They are all the results of the simulation experiment, readers can easily find out the distribution and tendency of result data. Grid lines of Figure 8~figure 10 use light gray color. Figure 11 shows the finally routes of local and global solutions of the experiment.
6. Please include the references I have suggested and also update more recent researches.
Answer: Thank you for supplementing valuable citation papers for us, we have added the two citations as [1] and [2] and used the two citations in the context. Also more relative references are added.
7. References in the text and references list are particularly not followed by the journal guidelines and please include DOI if they have.
Answer: We have examined all the references and the new supplemented ones, and modify all the improper places. Thank you for reminding us.

Round 2
Reviewer 1 Report
Authors have successfully addressed the suggested comments.